# Improvement of the SWAT model for event-based flood simulation on a sub-daily time scale

Dan Yu[1], Ping Xie[1,2], Xiaohua Dong[3,4], Xiaonong Hu[5], Ji Liu[3,4], Yinghai Li[3,4], Tao Peng[3,4], Haibo Ma[3,4], Kai Wang[6], Shijin Xu[6]

[1]State Key Laboratory of Water Resources and Hydropower Engineering Science, Wuhan University, Wuhan, 430072, China
[2]Collaborative Innovation Center for Territorial Sovereignty and Maritime Rights, Wuhan, 430072, China
[3]College of Hydraulic & Environmental Engineering, China Three Gorges University, Yichang, 443002, China
[4]Hubei Provincial Collaborative Innovation Center for Water Security, Wuhan, 430070, China
[5]Institute of Groundwater and Earth Sciences, Jinan University, Guangzhou 510632, China
[6]Hydrologic Bureau of Huaihe River Commission, Bengbu, 233001, China

*Correspondence to*: Xiaohua Dong (xhdong@ctgu.edu.cn)

**Abstract.** Flooding represents one of the most severe natural disasters threatening the development of human society. A model that is capable of predicting the hydrological responses in watershed with management practices during flood period would be a crucial tool for pre-assessment of flood reduction measures. The Soil and Water Assessment Tool (SWAT) is a semi-distributed hydrological model that is well capable of runoff and water quality modeling under changed scenarios. The original SWAT model is a long-term yield model. However, a daily simulation time step and a continuous time marching limit the application of the SWAT model for detailed, event-based flood simulation. In addition, SWAT uses a basin level parameter that is fixed for the whole catchment to parameterize the Unit Hydrograph (UH), thereby ignoring the spatial heterogeneity among the sub-basins when adjusting the shape of the UHs. This paper developed a method to perform event-based flood simulation on a sub-daily time scale based on SWAT2005 and simultaneously improved the UH method used in the original SWAT model. First, model programs for surface runoff and water routing were modified to a sub-daily time scale. Subsequently, the entire loop structure was broken into discrete flood events in order to obtain a SWAT-EVENT model in which antecedent soil moisture and antecedent reach storage could be obtained from daily simulations of the original SWAT model. Finally, the original lumped UH parameter was refined into a set of distributed ones to reflect the spatial variability of the studied area. The modified SWAT-EVENT model was used in the Wangjiaba catchment located in the upper reaches of the Huaihe River in China. Daily calibration and validation procedures were first performed for the SWAT model with long-term flow data from 1990 to 2010, after which sub-daily ( $\Delta t = 2$ h ) calibration and validation in the SWAT-EVENT model were conducted with 24 flood events originating primarily during the flood seasons within the same time span. Daily simulation results demonstrated that the SWAT model could yield very good performances in reproducing streamflow for both whole year and flood period. Event-based flood simulation results simulated by the sub-daily SWAT-EVENT model indicated reliable performances, with $E_{NS}$ values varying from 0.67 to 0.95. The SWAT-EVENT model, compared to the SWAT model, particularly improved the simulation accuracies of the flood peaks. Furthermore, the SWAT-EVENT model results of the two

UH parameterization methods indicated that the use of the distributed parameters resulted in a more reasonable UH characterization and better model fit compared to the lumped UH parameter.

Keywords: SWAT model; Event-based flood simulation; Antecedent conditions; Unit Hydrograph

## 1 Introduction

A flood represents one of the most severe natural disasters in the world. It has been reported that nearly 40 % of losses originating from natural catastrophes are caused by floods (Adams Iii and Pagano, 2016). Floods have caused enormous losses to economy, society and ecological environment around the world (Doocy et al., 2013;Werritty et al., 2007;Guan et al., 2015). China is a flood-prone country, which suffers from severe flooding almost every year (Zhang et al., 2002). In this situation, protection against flooding has always been the government's primary task that brooks no delay. A series of structural and non-structural flood mitigation measures have been conducted to control and manage the floods (Guo et al., 2018). However, accurate flood simulations would be particularly important for such design- or management-related issues.

Numerous hydrological models have been developed since their first appearance. According to the spatial discretization method, these existing hydrological models can be divided into two categories: lumped models and distributed (semi-distributed) models (Maidment, 1994). Although lumped models are generally accepted for flood forecast and simulation due to the structural simplicity, computational efficiency and lower data requirements, they are not applicable to complex catchments since they do not account for the heterogeneity of the catchments (Yao et al., 1998;Hapuarachchi et al., 2011). Meanwhile, distributed (semi-distributed) models subdivide the entire catchment into a number of smaller heterogeneous sub-units with dissimilar attributes. It is the advantage for distributed (semi-distributed) models to incorporate the spatial characteristics of catchment such as land cover, soil properties, topography and meteorology (Yang et al., 2004;Yang et al., 2001). A large number of distributed or semi-distributed hydrological models have been applied in flood simulation. Beven et al. (1984) firstly tested the applicability of the TOPMODEL in flood simulation for three U.K. catchments and suggested that the model could be a useful approach for ungauged catchments. Variable Infiltration Capacity (VIC) model is also playing an increasing role in flood simulation (Wu et al., 2014;Yigzaw and Hossain, 2012). The applications of the HBV model for flood simulation could be found in many studies (Haggstrom et al., 1990;Grillakis et al., 2010;Kobold and Brilly, 2006). HEC-HMS model was able to provide reasonable flood simulation results in the San Antonio River Basin (Ramly and Tahir, 2016). Among many distributed (semi-distributed) models, the one that is capable of predicting the hydrological responses in watersheds with management practices would provide scientific reference for preventing flood and mitigating its adverse effects.

Soil and Water Assessment Tool (SWAT) model (Arnold et al., 1998) is a typical semi-distributed hydrological model that delineates a catchment into a number of sub-basins, which were subsequently divided into Hydrologic Response Units (HRU) representing the unique combination of land cover, soil type, and slope class within a sub-basin. SWAT model integrates well with Geographic Information System (GIS), having great potential in dealing with spatial flood control measures. In addition,

the SWAT model is widely applied for runoff and water quality modeling under changed scenarios (Glavan et al., 2015;Yu et al., 2018;Qiu et al., 2017;Baker and Miller, 2013;Yan et al., 2013).

SWAT is a continuous (i.e., long-term) model with a limited applicability toward simulating instantaneous hydrologic responses. Therefore, Jeong et al. (2010) extended the capability of SWAT to simulate operational sub-daily or even sub-hourly hydrological processes, the modifications of which primarily focused on the model algorithms to enable the SWAT model to operate at a finer time scale with a continuous modeling loop. Constrained by data availability in China (MWR, 2009), rainfall and discharge observations at a sub-daily time scale are usually collected during flood period, while daily data are measured otherwise. In this respect, hydrological models are usually applied at different time scales (i.e., a daily time scale for continuous simulations and a sub-daily time scale for event-based flood simulation) according to the availability of observed rainfall and discharge data (Yao et al., 2014a). Hence, a major constraint for the application of the SWAT model as modified by Jeong et al. (2010) is the conflict between a continuous simulation loop and the discontinuous observed sub-daily data in China.

To capture the sophisticated characteristics of flood events at a sub-daily time scale, a refinement of the spatial representation within the SWAT model is necessary. A dimensionless Unit Hydrograph (UH), which was distributed as a triangular shape and embedded within an sub-daily overland flow routing process in the SWAT model, was applied to relate hydrologic responses to specific catchment characteristics, such as the dimensions of the main stream and basin area, through applications of GIS or Remote Sensing (RS) software (Jena and Tiwari, 2006). Due to the spatial discretization in the SWAT model, the model parameters are grouped into three levels: (1) basin level parameters are fixed for the whole catchment; (2) sub-basin level parameters are varied with sub-basins; (3) HRU level parameters are distributed in different HRUs. By default, the UH-specific parameter in the SWAT model is programmed on the basin level, which means that spatial variation within a catchment is disregarded when adjusting the shape of the UH in each sub-basin. Given the spatial heterogeneity of the catchment, the application of this basin level adjustment parameter seems to be rather unconvincing. Moreover, because a great deal of research has primarily focused on daily, monthly or yearly simulations using the SWAT model, little effort has actually been provided toward demonstrating the usage of the UH method in the SWAT model.

This study developed a method to perform event-based flood simulation on a sub-daily time scale based on the SWAT model and simultaneously improved the UH method used in the original SWAT model in the upper reaches of the Huaihe River in China. SWAT is an open-source code model, which makes it possible to produce such a modification. The source code of SWAT2005 has an internal auto-calibration module and such integrated design of model simulation and auto-calibration is easily manageable and modified since there is no need to couple external optimization algorithms. The accessible SWAT2009 (rev. 528) and SWAT2012 (rev. 664) have removed auto-calibration routines, however, an independent program SWAT-CUP (Abbaspour et al., 2007) is provided instead. Admittedly, many improvements have been made from the SWAT2005 to the latest SWAT2012. According to the SWAT model updates in Seo et al. (2014), the major enhancements focused on the water quality modeling components, whereas the runoff modeling components in new SWAT versions were not so far different from those in the SWAT2005. This study was specific to the model modifications in runoff simulation, thus, the SWAT2005 was

considered to be appropriate. There are some other model modification studies (Dechmi et al., 2012;Jeong et al., 2010) based on the SWAT2005 version.

## 2 Study area and data

### 2.1 Study area

The Huaihe River basin (30°55'–36 °36' N, 111°55'–121°25' E) is situated in the eastern part of China. The Wangjiaba (WJB) catchment is situated within the upper reaches of the Huaihe River basin and was chosen as the study area for this paper (see Fig. 1). The WJB catchment has a drainage area of 30630 km$^2$, wherein the long channel reaches from the source region to the WJB outlet. The southwestern upstream catchment is characterized as a mountain range with a maximum elevation of 1110 m above sea level. The central and eastern downstream regions are dominated by plains. The study catchment is a subtropical

zone with an annual average temperature of 15 °C. The long-term average annual rainfall varies from 800 mm in the north to 1200 mm in the south. Since the catchment is dominated by a monsoon climate, approximately 60 % of the annual rainfall is received during the flood season ranging from mid-May to mid-October. Severe rainfall events within the study area typically transpire during the summer, frequently resulting in severe floods (Zhao et al., 2011).

### 2.2 Model dataset

To construct and execute the SWAT model, a Digital Elevation Model (DEM), together with land use and soil type data, is required. Climate data, including that of rainfall, temperature, wind speed, etc., are also used. Table 1 lists the model data used in this study.

The DEM data in this study were downloaded from the website of the U.S. Geological Survey (USGS) with a spatial resolution of 90 m. The study catchment was divided into 136 sub-basins according to the catchment delineation, as shown in Fig. 1.

A land use map was produced from the Global Land Cover 2000 (GLC2000) data product with a grid size of 1 km (Bartholomé and Belward, 2005). Six categories of land use were identified for this catchment: agricultural land (80.51 %), forest-deciduous (6.76 %), forest-evergreen (2.26 %), range-brush (1.09 %), range-grasses (8.09 %) and water (1.29 %).

Soil data were obtained from the Harmonized World Soil Database (HWSD) with a spatial resolution of 30 arc-seconds. The HWSD also provides an attributed database that contains the physico-chemical characteristics of soil data worldwide

(Nachtergaele et al., 2012). Since the built-in soil database within the SWAT model does not cover the study area, additional soil parameters were calculated using the method proposed by Jiang et al. (2014). Soil reclassification in the study area was in accordance with the FAO-90 soil system. Consequently, Eutric Planosols and Cumulic Anthrosols are the two main soil types with area percentages of 24.71 % and 19.95 %, respectively.

The SWAT model has developed a weather generator (WXGEN) to fill the missing climate data by the use of monthly statistics.

Relative humidity, wind speed, solar radiation and the minimum and maximum air temperatures were obtained from the

Climate Forecast System Reanalysis (CFSR), which was designed based on the forecast system of the National Centers for Atmospheric Prediction (NCEP) to provide estimation for a set of climate variability from 1979 to the present day. There were 30 weather stations included in the study catchment.

A dense rain gauge network consisting of 138 gauges is distributed throughout the study area as illustrated in Fig. 1. By default, SWAT structure allows only one rainfall input for each delineated sub-basin. Thus, sub-basins without available rainfall gauge would be automatically assigned the nearest one. For sub-basins with multiple rainfall gauges, Thiessen polygon method (Thiessen, 1911) was utilized to derive the rainfall input. Rainfall is the main driving force for hydrological models, and therefore accurate representation of spatially distributed rainfall is essential in hydrological modeling. Cho et al. (2009) compared three different methods to incorporate spatially variable rainfall into the SWAT model and recommended the Thiessen polygon approach in catchments with high spatial variability of rainfall due to its robustness to catchment delineation. Daily observed rainfall data were retrieved from 1991 to 2010 with coverage during the entire year, while sub-daily ( $\Delta t = 2$ h ) rainfall data are only available for several flood events from May to September within the same time span.

## 3 Methodologies

### 3.1 Development of a sub-daily event-based SWAT model

The original SWAT model was designed for continuous simulations using a daily time step. The SWAT model operates most effectively during the prediction of long-term hydrological responses to land cover changes or soil management practices with daily time step (Jeong et al., 2011). When faced with flood simulation issues, a finer time scale is required to realistically capture the instantaneous changes representative of flood processes.

Therefore, the original daily simulation-based SWAT model first needs to be modified in order to perform sub-daily simulations. In a previous study, the sub-daily and even the sub-hourly modeling capacities of the SWAT model have been developed to allow flow simulations with any time step less than a day (Jeong et al., 2010). In the original SWAT model, the surface runoff lag was estimated by a first order lag equation, which was represented by a function of the concentration time and the lag parameter. However, this lag equation was implicitly fixed with daily time interval. Jeong et al. (2010) then introduced the simulation time interval into the lag equation to lag a fraction of the surface runoff at the end of each time step. In addition, channel and impoundment routings were also estimated at operational time interval while other processes such as base flow and evapotranspiration were calculated by equally dividing the daily results over the time steps. In this study, the modifications from daily modeling to sub-daily modeling followed the methods proposed by Jeong et al. (2010). Second, the modified sub-daily SWAT model must be applied in such a manner to achieve the simulation of individual flooding events rather than to simulate in a continuous way, as performed in the original SWAT model. Event-based sub-daily flood modeling is necessary for these reasons: (1) to enable the modelers to acknowledge the detailed information of up-coming floods and (2) to potentially conduct flood simulation within a watershed without possessing continuously recorded hydrologic data at short

time step. To enable the SWAT model to simulate individual flood events, the original source codes were modified and compiled into a new version known as SWAT-EVENT. In the source code of SWAT2005, the subroutine "simulate" contains the loops governing the hydrological processes following the temporal marching during the entire simulation period. Here, the continuous yearly loop was set into several flood events, meanwhile, the continuous daily loop was broken into flood events according to the specific starting and ending dates.

However, the event-based modeling requires a separate method to derive the antecedent conditions of model states. The combination of daily continuous modeling and sub-daily event-based modeling was used in this study (Fig. 2). A continuous daily rainfall sequence was imported into the original SWAT model to independently perform long-term daily simulations. In the SWAT model, there are another two subroutines "varinit" and "rchinit" initializing the daily simulation variables for the land phase of the hydrologic cycle and the channel routing, respectively. In the SWAT-EVENT model, condition judgments were added into those two initialization subroutines. That is, when the simulation process is at the beginning of a given flood event, antecedent soil moisture and antecedent reach storage are set equal to the respective values extracted from the long-term daily simulations of the original SWAT model; otherwise, they should be updated by the SWAT-EVENT model simulation states of the previous day.

## 3.2 Application of Unit Hydrographs with distributed parameters

The dimensionless UH method employed in the SWAT model exhibits a triangular shape (SCS, 1972), as shown in Fig. 3, wherein the time $t$ (h) represents the X-axis, and the ratio of the discharge to peak discharge represents the Y-axis. This UH is defined as follows:

$$q_{uh} = \frac{t}{t_p} \quad \text{if } t \le t_p$$
$$q_{uh} = \frac{t_b - t}{t_b - t_p} \quad \text{if } t > t_p \tag{1}$$

where $q_{uh}$ is the unit discharge at time $t$, $t_p$ is the time to the peak (h), and $t_b$ is the time base (h). Then, the dimensionless UH is expressed by dividing by the area enclosed by the triangle (Jeong et al., 2010). There are two time factors determining the shape of the triangular UH, which are defined by the following equations:

$$t_b = 0.5 + 0.6 \bullet t_c + t_{adj} \tag{2}$$

$$t_p = 0.375 \bullet t_b \tag{3}$$

where $t_c$ is the concentration time for the sub-basin (h), and $t_{adj}$ is a shape adjustment factor for the UH (h) (Neitsch et al., 2011).

The time of concentration $t_c$ can be calculated based upon the geographic characteristics of the sub-basin considered, for which $t_c$ is denoted by the accumulation of the overland flow time $t_{ov}$ (h) and the channel flow time $t_{ch}$ (h):

$$t_c = t_{ov} + t_{ch} \tag{4}$$

$$t_{ov} = \frac{L_{slp}^{0.6} \bullet n^{0.6}}{18 \bullet S_{sub}^{0.3}} \tag{5}$$

$$t_{ch} = \frac{0.62 \bullet L \bullet n^{0.75}}{A^{0.125} \bullet S_{ch}^{0.375}} \tag{6}$$

where $L_{slp}$ is the average slope length for the sub-basin under consideration (m); $n$ is the Manning coefficient for the sub-basin; $S_{sub}$ is the average slope steepness of the sub-basin (m m$^{-1}$); $L$ is the longest tributary length in the sub-basin (km); $A$ denotes the area of the sub-basin (km$^2$); and $S_{ch}$ is the average slope of the tributary channels within the sub-basin (m m$^{-1}$). According to catchment discretization, Table 2 appears obvious spatial differences of the geographical attributes among sub-basins. For instance, the values of sub-basin area $A$ vary from 0.09 km$^2$ to 879.16 km$^2$ with a Coefficient of Variation (CV) of 0.74. The average slope of the sub-basin $S_{sub}$ and the average slope of the tributary channels $S_{ch}$ are topographic-related parameters, showing much higher values in source sub-basins than those in downstream sub-basins. Spatially, the CV values of $S_{sub}$ and $S_{ch}$ in Table 2 are 1.28 and 1.18. As a result, the overland flow time $t_{ov}$ and the channel flow time $t_{ch}$ affected by all those geographical attributes are non-homogeneous in the spatial distribution, especially for the $t_{ch}$ with the CV value of 0.91. Since the channel flow time $t_{ch}$ dominates the concentration time $t_c$, the CV of $t_c$ is 0.81 in Table 2. According to Eq. (2), the time base of the UH ($t_b$) is determined by both concentration time for the sub-basin ($t_c$) and shape adjustment factor ($t_{adj}$) concurrently. However, the UH parameter $t_{adj}$ in Eq. (2) is a basin level parameter possessing a lumped value for all sub-basins, meaning that the spatial heterogeneity of $t_b$ may be homogenized. Hypothetically, the CV value of the $t_b$ would decrease from 0.72 to 0.09 along with the increase of UH parameter $t_{adj}$ from 0 h to 30 h in Fig. 4. Generally, the time base of triangular UH ($t_b$) should be reduced to produce increased peak flow for steep and small sub-basins, or should be increased to produce decreased peak flow for flat and large sub-basins. Thus, the shape adjustment parameter $t_{adj}$ was modified from the basin level to the sub-basin level, and renamed $t_{subadj}$ which allowed the UHs to be adjusted independently by distributed values.

### 3.3 Model calibration and validation

### 3.3.1 Sensitivity analysis

Sensitivity analysis is a process employed to identify parameters that significantly influence model performance (Holvoet et al., 2005). Generally, sensitivity analysis takes priority over the calibration process to reduce the complexity of the latter (Sudheer et al., 2011). Here, a combined Latin-Hypercube and One-factor-At-a-Time (LH-OAT) sampling method embedded within the SWAT model (Griensven et al., 2006) was used to conduct a sensitivity analysis. LH-OAT method firstly subdivides

each parameter into $N$ stratums with a probability of $1/N$. Sampling points are randomly generated so that one parameter is sampled only once at each strata. Then, the local sensitivity of a parameter at one sampling point is calculated as:

$$S_{ij} = 200 \bullet \left| \frac{\left[ y\left(\theta_1,...,\theta_i + \Delta_i,...,\theta_P\right) - y\left(\theta_1,...,\theta_P\right)\right] / \left[ y\left(\theta_1,...,\theta_i + \Delta_i,...,\theta_P\right) + y\left(\theta_1,...,\theta_P\right)\right]}{\Delta_i} \right| \tag{7}$$

where $S_{ij}$ is the partial effect of parameter $\theta_i$ at the LH sampling point $j$; $y$ is the model output (or objective function);

$\Delta_i$ is the perturbation of parameter $\theta_i$ and $P$ is the number of parameters. The final sensitivity index $S_i$ for the parameter

$\theta_i$ is derived by averaging these partial effects of each loop for all LH points (i.e., $N$ loops). The greater the $S_i$, the more sensitive the model response is to that particular parameter.

It is highly recommended to identify the model parameters that can represent the hydrological characteristics of specific catchment before blindly applying sensitivity analysis. Based on the reviews of the SWAT model applications (Griensven et al., 2006;Cibin et al., 2010;Roth and Lemann, 2016) and the analysis of the SWAT model parameters, a total of 16 parameters related to the streamflow simulation in study area were involved in sensitivity analysis (see Table 3) for daily simulation with the SWAT model. When it came to the event-based sub-daily flood simulation with SWAT-EVENT model, additional distributed UH parameter $t_{\text{subadj}}$ (i.e., a total of 17 model parameters) was also considered. For both models, the objective function $y$ in Eq. (7) represented the residual sum of squares of stream flow between the simulated set and the measured set. Specifically, sensitivity analysis of the SWAT model was conducted not only for long-term period, but also for the same flood period as the SWAT-EVENT simulation. According to the sensitivity ranks of $S_i$, the upper-middle ranking parameters would be used for the calibration procedure, while the values of the other parameters were set to their default values.

### 3.3.2 Daily calibration and validation with the SWAT model

Before effectively applying a hydrological model, a calibration process aims to estimate the model parameters that minimize the errors between the observed and simulated results is usually necessary. The Shuffled Complex Evolution (SCE-UA) algorithm (Duan et al., 1992) is a global optimization technique that is incorporated as a module into the SWAT model. The SCE-UA algorithm has been applied to multiple physically based hydrological models (Sorooshian et al., 1993;Luce and Cundy, 1994;Gan and Biftu, 1996) and has exhibited good performance similar to other global search procedures (Cooper et al., 1997;Thyer et al., 1999;Kuczera, 1997;Jeon et al., 2014).

Daily simulations were performed within the time span, from 1990 to 2010, using daily observed data at the outlet of WJB. During this phase, the SWAT model was also conducted in two ways, calibrating for long-term period and calibrating for flood period. For long-term period case, one year (1990) was selected as the model warm-up period, the period from 1991 to 2000 was used for the model calibration, and the remaining data from 2001 to 2010 were employed for validation. For flood period

calibrating, what was different was that the objective function only covered several flood events, which were consistent with the SWAT-EVENT application.

Multiple statistical values, including the Nash-Sutcliffe efficiency coefficient ($E_{NS}$) (Nash and Sutcliffe, 1970), ratio of the root mean square error to the standard deviation of measured data ($R_{SR}$) (Singh et al., 2005) and the percent bias ($P_{BIAS}$) (Gupta et al., 1999), were selected in this study to evaluate the daily model performances, as shown in Eq. (8), (9) and (10). The $E_{NS}$ provides a normalized statistic indicating how closely the observed and simulated data match with each other, wherein a value equal to 1 implies an optimal model performance insomuch that the simulated flow perfectly matches the observed flow. The $R_{SR}$ index standardizes the root mean square error using the observations standard deviation, varying from 0 to a positive value. The optimal value of $R_{SR}$ is 0, which indicates the perfect model simulation. The $P_{BIAS}$ detects the degree that the simulated data deviates from the observed data.

$$E_{NS} = 1 - \left[ \frac{\sum_{i=1}^{n}\left(Q_{obs}(i) - Q_{sim}(i)\right)^2}{\sum_{i=1}^{n}\left(Q_{obs}(i) - \overline{Q}_{obs}\right)^2} \right] \tag{8}$$

$$R_{SR} = \frac{\sqrt{\sum_{i=1}^{n}\left(Q_{obs}(i) - Q_{sim}(i)\right)^2}}{\sqrt{\sum_{i=1}^{n}\left(Q_{obs}(i) - \overline{Q_{obs}}\right)^2}} \tag{9}$$

$$P_{BIAS} = \left[ \frac{\sum_{i=1}^{n}\left(Q_{obs}(i) - Q_{sim}(i)\right)\cdot 100}{\sum_{i=1}^{n}Q_{obs}(i)} \right] \tag{10}$$

where $Q_{obs}(i)$ is the $i$ th observed streamflow (m$^3$ s$^{-1}$); $Q_{sim}(i)$ is the $i$ th simulated streamflow (m$^3$ s$^{-1}$); $n$ is the length of the time series.

### 3.3.3 Event-based sub-daily calibration and validation with the SWAT-EVENT model

In this study, the SWAT-EVENT model employed the same built-in automatic calibration subroutine as the SWAT model did. Sub-daily simulations with the SWAT-EVENT model were conducted within the same time span as the daily simulation, with a primary focus on the flood season with a series consisting of 24 flood events, two-thirds of which were utilized for the calibration while the rest were used for validation. Preferential implementation was applied to daily calibration from which the antecedent conditions were extracted.

$E_{NS}$, relative peak discharge error ($E_{RP}$), relative peak time error ($E_{RPT}$) and relative runoff volume error ($E_{RR}$) were selected as the performance evaluation statistics for the flood event simulations to comply with the Accuracy Standard for Hydrological

Forecasting in China (MWR, 2008). $E_{RP}$, $E_{RPT}$, and $E_{RR}$ are specific indicators used to indicate whether the accuracies of the simulations reach the national standard (MWR, 2008). They are considered to be sufficiently qualified when the absolute values are less than 20 %, 20 % and 30 %, respectively.

## 4 Results

### 4.1 Sensitivity analysis results

Sensitivity results for daily simulation with the SWAT model are listed in Table 3. The sensitivity rank for a single parameter shows tiny differences between the two types of analysis period for SWAT simulation, with the changes in all parameter ranks less than 3. According to a previous study (Cibin et al., 2010), the sensitivity of SWAT parameters was proved to vary in low, medium and high streamflow regimes. The long-term period analysis in Table 3 consists of different flow regimes, but presents almost the same sensitivity ranks as the flood period case, indicating that the high streamflow would dominate the sensitivity results in the long-term period analysis. The identified 7 sensitive parameters of the daily SWAT model cover multiple main hydrological processes, i.e. channel routing (CH_N2 and CH_K2), runoff (SURLAG and CN2), groundwater (ALPHA_BF), evaporation (ESCO) and soil water (SOL_AWC), not only for long-term period but also for flood period. According to Table 3, it is clear that both the year-round streamflow and the high streamflow are most sensitive to CH_N2 due to its top sensitivity rank.

Table 3 also presents the sensitivity results for event-based flood simulation with SWAT-EVENT model at sub-daily time scale. Sensitivity of some parameters differs widely from its performance in flood period analysis with SWAT model at daily time scale. The sensitivity ranks of BLAI, CH_K2, ESCO, SOL_K, and SURLAG have changed more than 5, which could be caused by the differences in hydrological simulation between the SWAT model and the SWAT-EVENT model. It is noteworthy that the UH parameter $t_{subadj}$, peculiar to the SWAT-EVENT model, can significantly influence the event-based flood simulation at sub-daily time scale with corresponding sensitivity ranking of 3 in Table 3. Though there exists differences among the daily SWAT model and the sub-daily SWAT-EVENT model, the same point is that the parameter CH_N2 is recognized as the most important parameters for both two models. In general, the top 8 sensitive parameters (ALPHA_BF, CH_N2, CN2, GWQMN, SOL_AWC, SOL_K, SOL_Z and $t_{subadj}$) are considered to influence the event-based sub-daily flood simulation significantly.

### 4.2 Daily simulation results

The final calibrated parameters for daily simulation with the SWAT model are presented in Table 4. The model performances for daily streamflow simulations at outlet WJB are summarized in Table 5. For long-term calibration, the $E_{NS}$ value is 0.76 for

the calibration period and 0.80 for the validation period. These two values of the daily $E_{NS}$ both exceed 0.75, which is considered to be "very good" according to performance ratings for evaluation statistics recommended by Moriasi et al. (2007). The daily $R_{SR}$ values are 0.49 and 0.44 for the calibration and validation, respectively, indicating that the root mean square error values are less than half the standard deviation of measured data, i.e. the "very good" model performances suggested by Moriasi et al. (2007). The SWAT model overestimates the streamflow by 5.72 % for calibration while underestimating the streamflow by 8.38 % for validation. The calculated results of $P_{BIAS}$ in Table 5 also attain the "very good" rating. Visual comparisons between the observed and simulated streamflows for both of the calibration and validation periods are shown in Fig. 5, from which it can be observed that the SWAT model could simulate well the temporal variation of long-term streamflow at daily time scale. In general, the daily simulation results obtained from the SWAT model at WJB demonstrate decent applicability and can consequently represent a preliminary basis for further flood event simulation.

When focusing on event period calibration and validation, all statistical criteria in Table 5 indicate high accuracy of the daily SWAT model for flood period simulation.

## 4.3 Event-based simulation results

Table 4 shows the optimum values of parameters used in the SWAT-EVENT model simulation. The sub-daily simulation results for 24 flood events, as shown in Table 6, exhibit reliable performances of the SWAT-EVENT model, with $E_{NS}$ values varying from 0.67 to 0.95. The qualified ratios of $E_{RP}$, $E_{RPT}$ and $E_{RR}$ are 75%, 95.8% and 91.6%, respectively. Meanwhile, observed and simulated sub-daily flood hydrographs are displayed in Fig. 6 and Fig. 7. It is clearly that the SWAT-EVENT model has the ability to accurately simulate the sub-daily flood events, except for the event 20020722. Moreover, for specific floods (i.e., 19960628, 19980725, 20050707 and 20070701), it is remarkable to see that the SWAT-EVENT model owns the outstanding performances in simulating flood events with multi peaks.

Table 6 also displays the model performances of the daily simulation results using the SWAT model specific for flood period. All daily $E_{NS}$ values are lower than the sub-daily ones, indicating that the flood hydrographs simulated by the sub-daily SWAT-EVENT model are much more reliable than those simulated by the daily SWAT model. In addition, the peak flows simulated by the SWAT-EVENT model on a sub-daily time scale are much closer to the observed flows relative to the predictions obtained from the SWAT model on a daily time scale, especially for flood events with high peak flows in Table 6. There are eight flood events (19910610, 19910629, 19960628, 20020622, 20030622, 20050707, 20050822 and 20070701) that exhibit peak flows greater than 5000 $m^3$ $s^{-1}$. The sub-daily simulation results of these eight floods were aggregated into daily averages and then compared with those of the daily simulations, the results of which are illustrated in Fig. 8. It can be concluded that the daily simulations are likely to miss the high flood peaks. The more effective performances of the SWAT-EVENT model could be due to rainfall data with a higher temporal resolution and the model calculation with more detailed time steps, which can capture the instantaneous changes representative of flood processes.

All statistical indicators suggest that the SWAT-EVENT model can accurately reproduce the dynamics of observed flood events based upon antecedent conditions extracted from SWAT daily simulations.

**4.4 Effects of the UH parameter level on SWAT-EVENT model performances**

To analyze the effects of the level of UH parameter on SWAT-EVENT model simulations, the default lumped UH parameter $t_{adj}$ was calibrated while the other parameters remained unchaged exactly as the sub-basin level case was calibrated in Table 4. The optimized basin level UH parameter ($t_{adj}$) displays a uniform value of 15.75 h for all sub-basins, while the sub-basin level UH parameters ($t_{subadj}$) are distributed in sub-basins, ranging from 4.81 h to 120.33 h. As a consequence, the optimized $t_{subadj}$ value enables the base time ($t_b$) and the peak time ($t_p$) of the UHs within the ranges of 6.13 h - 141.34 h and 2.30 h- 53.00 h, respectively. While for the basin level UH parameter case, the values of $t_b$ and $t_p$ distribute in a relatively narrow range, i.e. 17.07 h -36.76 h for $t_b$ and 6.40 h - 13.78 h for $t_p$. More of a concern, according to Fig. 4, is the CV value of $t_b$ or $t_p$ would be reduced to less than 0.2, meaning that the spatial heterogeneity of UH time factors is homogenized due to the constrains between sub-basins when adjusting the basin level UH parameter. As expected, the application of sub-basin level UH parameters would keep the CV value of $t_b$ or $t_p$ at 0.79, which corresponds quite closely to the CV value of $t_c$ in Table 2. Thus, the spatial inhomogeneity of geographical features can be better represented by the use of sub-basin level UH parameters.

The SWAT-EVENT simulation results using the basin level UH parameter are also presented in Table 4. Compared with the sub-basin level case, the basin level case induces significant decrease in the qualified ratio of $E_{RPT}$ from 95.8 % to 79.1 %. Intuitive comparisons for relative peak discharge error ($E_{RP}$) and relative peak time error ($E_{RPT}$) under both UH parameter levels could be found in Fig. 9. When simulating from sub-basin level UH case to basin level UH case, more than half of the total 24 flood events and nearly all the flood events show respectively increased peak discharge error and peak time error. Thus, it can be concluded that changing the spatial level of the UH parameter would affect the flood peak simulations significantly, especially for the peak time error. In this procedure, however, model parameters except for the UH parameter remain fixed, so it is not surprising that there is little change in the specific values of the relative runoff volume error ($E_{RR}$) between the two cases in Table 4. All these findings indicate that the application of sub-basin level UH parameters in the SWAT-EVENT model can improve the simulation accuracies of flood peaks.

The overall distributions of $E_{NS}$ statistics for flood events for the two UH methods (i.e., the basin level UH parameter vs. the sub-basin level UH parameters) are plotted in Fig. 10. The box plots therein exhibit rectangle heights equal to the interquartile range (IQR), the upper and lower ends of which are separately marked with the upper and lower quartile values, respectively. The median is represented by a line transecting either of the two rectangles. The extended whiskers denote the range of the batch data (Massart et al., 2005;Cox, 2009). According to Table 4 and Fig. 10, the SWAT-EVENT model using sub-basin

level UH parameters demonstrates improvements for event-based flood simulation. For the sub-basin level case in Fig. 10, half of the $E_{NS}$ values range from 0.83 (lower quartile) to 0.91 (upper quartile), with a median of 0.87, which can potentially represent the second flood forecasting accuracy standard (i.e. B) according to MWR (MWR, 2008). However, the basin level case performs comparatively poorly with regard to reproducing the flood hydrograph, wherein the majority of $E_{NS}$ values vary between 0.78 and 0.88. In comparison, the application of spatially distributed UH parameters allows the SWAT-EVENT model to simulate the flood events more accurately.

## 5 Discussion

### 5.1 Sub-daily simulation vs. daily simulation

Floods are always triggered by intense rainfall events with short duration. In order to adequately capture and analyze the rapid response of flood events, simulation time step at sub-daily resolution is preferred. Normally, an appropriate simulation time step is chosen depend on the catchment response time to a rainfall event. According to the catchment delineation and geographical features of sub-basins in Table 2, the general average concentration time of sub-basins is found to be less than 24 h. Moreover, considering the time interval of observed data acquisition (i.e. 2 h to 6 h), the 2-hour simulation step chosen in this study was more than sufficient for flood simulation. The remarkable performances of the sub-daily SWAT-EVENT model for peak flow simulations (as shown in Table 6 and Fig. 8) adequately confirmed the superiority of using sub-daily time step in simulating flood hydrographs. In this study, daily surface runoff was calculated using the SCS curve number method in the SWAT model, whereas sub-daily surface runoff was calculated using the Green & Ampt infiltration method in the SWAT-EVENT model. In terms of the comparison of these two methods, Vol. (1999) argued that the advantage of Green & Ampt method was the considerations of sub-daily rainfall intensity and duration, meanwhile, a rainstorm might not be fully represented by total daily rainfall used in SCS method due to its high variation in temporal distribution. Beyond that, as stated by Jeong et al. (2010), the physically based hydrological processes simulating at a short time scale would contribute to the reinforcement of model simulation accuracy.

### 5.2 Event-based simulation vs. continuous simulation

Pathiraja et al. (2012) may argue that the continuous simulation for design flood estimation was becoming increasingly important. Nevertheless, in operational flood simulation and prediction perspectives, many endusers and practitioners are still in favor of the event-based models (Coustau et al., 2012;Berthet et al., 2009). The emphasis on event-based modeling in this study was due to the unavailability of the long continuous hydrological data at sub-daily time scale. Such data scarcity issue has also promoted the applications of the event-based models in some developing countries (Hughes, 2011;Tramblay et al.,

2012). More broadly, the preferred event-based approach is highlighted when the hydrological model is used for investigating the effect of heavy rainfall on environmental problems such as soil erosion and contaminant transport (Maneta et al., 2007). Several studies have declared that the catchment's antecedent moisture conditions prior to a flood event can have a strong influence on flood responses, including the flood volume, flood peak flow and its duration (Rodrã-Guez-Blanco et al., 2012;Tramblay et al., 2012;Coustau et al., 2012). However, the major drawback of event-based models lies in its initialization: external information is needed to set the antecedent conditions of a catchment (Berthet et al., 2009;Tramblay et al., 2012). To address the initialization issue, some efforts have been placed to set up the initial conditions of event-based models, such as in-situ soil moisture measurements, retrieved soil moisture from the remote sensing products and continuous soil moisture modeling. Among these methods, continuous soil moisture modeling using the daily data series to estimate sub-daily initial conditions would be a traditional solution, as suggested by Nalbantis (1995). Tramblay et al. (2012) also tested different estimations of the antecedent moisture conditions of the catchment for an event-based hydrological model and concluded that the continuous daily soil moisture accounting method performed the best. However, there might be some deficiencies in the continuous simulation of the SWAT model in this study. On the one hand, the continuous soil moisture modeling required long data series and took a long time to implement. On the other hand, the continuous SWAT model was calibrated using the sum of squares of the residuals as the objective function, which was more sensitive to high flows than low flows. As a consequence, the SWAT model ensured the simulation accuracy at the expense of the low flow performances, which would certainly bring errors to the estimations of antecedent moisture conditions. As Coustau et al. (2012) declared, event-based models were very convenient for operational purposes, if the initial wetness state of the catchment would be known with good accuracy. Although the continuous modeling approach used in this study was not the perfect solution for the determination of the catchment antecedent conditions, it was still an effective method as the preliminary preparation for the simulation of the SWAT-EVENT model due to the good goodness-of-fit in Fig.6 and Fig.7. Since the goal of this research was to ascertain the applicability of the newly developed SWAT-EVENT model on event-based flood simulation, it was accepted to have a lower performance in calculating the antecedent conditions. Active microwave remote sensing has proved the feasibility and rationality of obtaining temporal and spatial soil moisture data. It means that there is a potential interest of using the remote sensing data to estimate the initial conditions (Tramblay et al., 2012).

## 5.3 Distributed UH parameters vs. lumped UH parameter

The UH method is used to spread the net rainfall over time and space, representing the most widely practiced technique for determining flood hydrographs. The main difference between the two applications of the UH parameter is, in essence, the method for surface runoff routing within the sub-basins. The application of the sub-basin level UH parameters allowed distributed parameter value for each sub-basin, while the basin level UH parameter application consistently applied a lumped value for all sub-basins. All but the derived UH shape of the distributed UH case was identical to these of the lumped UH case. Therefore, the difference in the simulations of the two UH parameter cases resulted from the surface runoff routing method.

As seen from the aforementioned model performance assessment in Table 6 and Fig. 9, the capability of the SWAT-EVENT model with basin-level UH parameter for event-based flood simulation was downgraded relative to the sub-basin level case. It is known that Sherman (1932) first proposed the UH concept in 1932. However, because the UH proposed by Sherman is based on observed rainfall-runoff data at gauging sites for hydrograph derivations, it is only applicable for gauged basins (Jena and Tiwari, 2006). A prominent lack of observed data promoted the appearance of the Synthetic Unit Hydrograph (SUH), which extended the application of the UH technique to ungauged catchments. The triangular dimensionless UH used in this study denotes the traditional derivation of SUHs, which relates hydrologic responses to the catchment geographic characteristics according to Eq. (2) - Eq. (6). Therefore, it can be inferred that the shape feature of the UH should be region-dependent. A lumped UH parameter used for the whole catchment would lead to either sharpening the peak flows in large sub-basins, or flattening the peak flows in small sub-basins. On the whole, hydrological behaviors among sub-basins would tend to be homogenized. As indicated in Table 6, Fig. 9 and Fig. 10, there was a positive effect from the application of the distributed UH parameters on flood simulation.

In addition to the triangular dimensionless UH used in this study, there are many other available methods for derivation of the SUH. Bhunya et al. (2007) compared four probability distribution functions (pdfs) in developing SUH and concluded that such statistical distributions method performed better than the traditional synthetic methods. Furthermore, the instantaneous unit hydrograph (IUH) is more capable of mathematically expressing the effective rainfall hyetograph and direct runoff hydrograph relationship in a catchment (Jeng and Coon, 2003). And Yao et al. (2014b) improved the flood prediction performance of the Xinanjiang model by the coupling of the geomorphologic instantaneous unit hydrograph (GIUH). Khaleghi et al. (2011) compared the accuracy and reliability of different UH methods and confirmed the high efficiency of the GIUH for flood simulation. There might be room for further improving the current UH method used in the SWAT-EVENT model.

## 6 Conclusions

The original SWAT model was not competent to flood simulation due to its initial design of long-term simulations with daily time-steps. This paper mainly focused on the modification of the structure of the original SWAT model to perform event-based flood simulation, which was applicable for the area without continuous long-term observations. The newly developed SWA-EVENT model was applied in the upper reaches of the Huaihe River. Model calibration and validation were made by the using of historical flood events, showing good simulation accuracy. To improve the spatial representation of the SWA-EVENT, the lumped UH parameters were then adjusted to the distributed ones. Calibration and validation results revealed the improvement of event-based simulation performances, especially for the flood peak simulation. This study expands the application of the original SWAT model in event-based flood simulation. Event-based runoff quantity and quality modeling has become a challenge task since the impact of hydrological extremes on the water quality is particularly important. The improvement of the SWAT model for event-based flood simulation in this study will lay the foundation for dealing with the event-based water quality issues.

The optimal parameters of the SWAT-EVENT model were obtained by the automatic parameter calibration module that integrated SCE-UA algorithm in this study. However, several factors such as interactions among model parameters, complexities of spatio-temporal scales and statistical features of model residuals may lead to the parameter non-uniqueness, which is the source of the uncertainty in the estimated parameters. Uncertainty of model parameters will be finally passed to the model results, hence leading to certain risks in flood simulation. In the future, emphasis will be placed on the quantification of the parameter uncertainty to provide better supports for flood operations.

**Data availability**

The DEM data were downloaded from the website http://srtm.csi.cgiar.org/.

The land use data (GLC2000) were downloaded from the website http://www.landcover.org/.

The soil data (HWSD) were downloaded from the website http://webarchive.iiasa.ac.at/Research/LUC/External-World-soil-database/HTML/.

The global weather data were downloaded from the website https://globalweather.tamu.edu/.

The rainfall observations at 138 stations and the discharge observations at the outlet (WJB) were provided by Hydrologic Bureau of Huaihe River Commission.

The source codes of SWAT model are available at the website http://swat.tamu.edu/.

**Competing interests**

The authors declare that they have no conflict of interest.

**Acknowledgments**

This research has been supported by Non-profit Industry Financial Program of Ministry of Water Resources of China (No. 201301066), National key research and development program (2016YFC0402700), the National Natural Science Foundation of China (No. 91547205, 51579181, 51409152, 41101511, 40701024), and Hubei Provincial Collaborative Innovation Center for Water Security.

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

**Table 1 SWAT model input data and sources for the Wangjiaba (WJB) catchment.**

| Data type | Resolution | Source | Description |
|---|---|---|---|
| DEM | 90m×90m | http://srtm.csi.cgiar.org/ | Digital Elevation Model |
| Land use | 1km×1km | http://www.landcover.org/ | Land use classification |
| Soil | 30 arc-second | http://www.fao.org/soils-portal/soil-survey/soil-maps-and-databases/harmonized-world-soil-database-v12/en/ | Soil type classification and characterization of soil parameters |
| Global weather data | 30 stations | https://globalweather.tamu.edu/ | Relative humidity, wind speed, solar radiation and the minimum and maximum air temperatures |
| Observed rainfall | 138 gauges | Hydrologic Bureau of Huaihe River Commission | Daily data: 1991-2010; sub-daily data: flood period during 1991-2010 |
| Observed streamflow | 1 gauges | Hydrologic Bureau of Huaihe River Commission | Wangjiaba station, daily data for 1991-2010, sub-daily data for flood period during 1991-2010 |

**Table 2 Geographic features of sub-basins for the Wangjiaba (WJB) catchment.**

| | $L_{slp}$ (m) | $S_{sub}$ (m/m) | $L$ (km) | $A$ (km²) | $S_{ch}$ (m/m) | $t_{ov}$ (h) | $t_{ch}$ (h) | $t_c$ (h) |
|---|---|---|---|---|---|---|---|---|
| Minimum | 28.46 | 0.01 | 0.71 | 0.09 | 0.000 | 0.14 | 0.13 | 1.37 |
| Maximum | 121.95 | 0.22 | 96.83 | 879.16 | 0.024 | 2.42 | 33.06 | 34.18 |
| Average | 100.42 | 0.04 | 37.44 | 221.88 | 0.005 | 0.91 | 6.03 | 6.94 |
| CV | 0.29 | 1.28 | 0.52 | 0.74 | 1.18 | 0.37 | 0.91 | 0.81 |

5    **Table 3 Parameters and parameter ranges used in sensitivity analysis and the final ranks of sensitivity analysis results.**

| Parameter | Definition | Lower bound | Upper bound | Daily simulation with SWAT model | | Event-based sub-daily simulation with SWAT-EVENT model |
|---|---|---|---|---|---|---|
| | | | | Long-term period | Flood period | |
| ALPHA_BF | Baseflow alpha factor (days). | 0 | 1 | 4 | 3 | 4 |
| BLAI | Maximum potential leaf area index. | 0 | 1 | 10 | 8 | 15 |
| CANMX | Maximum canopy storage (mm). | 0 | 10 | 11 | 11 | 12 |
| CH_K2 | Effective hydraulic conductivity in main channel alluvium (mm/hr). | 0 | 150 | 5 | 5 | 11 |
| CH_N2 | Manning's "n" value for the main channel. | 0.01 | 0.3 | 1 | 1 | 1 |
| CN2 [a] | Initial SCS runoff curve number for moisture condition II. | -25 | 25 | 3 | 4 | 2 |
| EPCO | Plant uptake compensation factor. | 0 | 1 | 12 | 12 | 16 |
| ESCO | Soil evaporation compensation factor | 0 | 1 | 6 | 6 | 17 |

| Parameter | Description | | | | | |
|---|---|---|---|---|---|---|
| GW_DELAY | Groundwater delay time (days). | 0 | 20 | 15 | 13 | 10 |
| GW_REVAP [b] | Groundwater "revap" coefficient. | -0.036 | 0.036 | 14 | 14 | 14 |
| GWQMN | Threshold depth of water in the shallow aquifer required for return flow to occur (mm). | 0.01 | 100 | 8 | 9 | 7 |
| REVAPMN [b] | Threshold depth of water in the shallow aquifer for "revap" or percolation to the deep aquifer to occur (mm). | -100 | 100 | 16 | 16 | 13 |
| SOL_AWC [a] | Available water capacity of the soil layer (mm /mm ). | -30 | 30 | 7 | 7 | 5 |
| SOL_K [a] | Saturated hydraulic conductivity (mm/hr). | -50 | 50 | 13 | 15 | 8 |
| SOL_Z [a] | Depth from soil surface to bottom of layer (mm). | -30 | 30 | 9 | 10 | 6 |
| SURLAG | Surface runoff lag coefficient. | 0 | 20 | 2 | 2 | 9 |
| $t_{subadj}$ [a] | Sub-basin level UH parameter (h) | -50 | 50 | | | 3 |

[a] These parameters are varied by multiplying a ratio (%) within the range.

[b] These parameters are varied by adding or subtracting a value within the range.

**Table 4 Calibrated parameter values for the SWAT model and the SWAT-EVENT model.**

| | Daily simulation with the SWAT model | | Event-based simulation with the SWAT-EVENT model | |
|---|---|---|---|---|
| Parameter | Value for long-term period calibrating | Value for flood period calibrating | Parameter | Value |
| CH_N2 | 0.10 | 0.19 | CH_N2 | 0.03 |
| SURLAG | 1.84 | 2.40 | CN2 | 24.60 |
| CN2 | 15.98 | 20.68 | $t_{subadj}$ | -10.40 |
| ALPHA_BF | 0.84 | 0.75 | SOL_Z | -7.91 |
| CH_K2 | 109.90 | 54.00 | GWQMN | 0.28 |
| ESCO | 0.94 | 1.00 | SOL_AWC | -29.71 |
| SOL_AWC | -18.01 | -9.26 | ALPHA_BF | 0.88 |
| | | | SOL_K | -48.84 |

**Table 5 SWAT model performance statistics for long-term period calibrating and flood period calibrating..**

| Statistical indicator | Long-term period calibrating | | Flood period calibrating | |
|:---:|:---:|:---:|:---:|:---:|
| | Calibration | Validation | Calibration | Validation |
| $E_{NS}$ | 0.76 | 0.80 | 0.78 | 0.81 |
| $R_{SR}$ | 0.49 | 0.44 | 0.48 | 0.44 |
| $P_{BIAS}$ (%) | 5.72 | -8.38 | 5.27 | -6.10 |

**Table 6 Performance evaluations for the daily SWAT model calibrating only for flood period, and the sub-daily SWAT-EVENT model performances with sub-basin level UH parameters and basin level UH parameter.**

| | Flood event | Start date | End date | Observed peak flow | Daily SWAT model simulation | | Sub-daily SWAT-EVENT model simulation with sub-basin level UH parameter | | | | | Sub-daily SWAT-EVENT model simulation with basin level UH parameter | | | | |
|---|---|---|---|---|---|---|---|---|---|---|---|---|---|---|---|---|
| | | | | | Simulated peak flow | $E_{NS}$ | Simulated peak flow | $E_{RP}$ | $E_{RPT}$ | $E_{RR}$ | $E_{NS}$ | Simulated peak flow | $E_{RP}$ | $E_{RPT}$ | $E_{RR}$ | $E_{NS}$ |
| | | | | (m³ s⁻¹) | (m³ s⁻¹) | | (m³ s⁻¹) | (%) | (%) | (%) | | (m³ s⁻¹) | (%) | (%) | (%) | |
| Calibration | 19910521 | 21-May | 10-Jun | 2935 | 2160 | 0.68 | 2740 | -6.64 | -6.04 | 13.38 | 0.93 | 2880 | -1.87 | -10.74 | 13.51 | 0.83 |
| | 19910610 | 10-Jun | 29-Jun | 7577 | 4500 | 0.80 | 7050 | -6.96 | 0.00 | -23.08 | 0.91 | 6860 | -9.46 | -9.46 | -22.98 | 0.90 |
| | 19910629 | 29-Jun | 21-Jul | 5931 | 3720 | 0.83 | 4960 | -16.37 | 0.88 | -31.25 | 0.85 | 4760 | -19.74 | -7.89 | -31.21 | 0.85 |
| | 19910804 | 4-Aug | 17-Aug | 4824 | 2830 | 0.70 | 3250 | -32.63 | 1.59 | -28.10 | 0.81 | 3030 | -37.19 | -1.59 | -28.05 | 0.80 |
| | 19950707 | 7-Jul | 18-Jul | 2613 | 1990 | 0.67 | 2310 | -11.60 | 2.44 | 7.77 | 0.89 | 2200 | -15.81 | -7.32 | 7.83 | 0.84 |
| | 19950803 | 3-Aug | 6-Sep | 922.1 | 1000 | 0.70 | 819 | -11.18 | -4.88 | -0.79 | 0.88 | 1220 | 32.31 | -5.49 | -0.78 | 0.78 |
| | 19960628 | 28-Jun | 25-Jul | 5298 | 3170 | 0.36 | 5200 | -1.85 | -4.20 | 0.54 | 0.67 | 5750 | 8.53 | -5.34 | 0.74 | 0.53 |
| | 19960917 | 17-Sep | 26-Sep | 1239 | 1550 | 0.86 | 1520 | 22.68 | 7.32 | 11.04 | 0.82 | 1340 | 8.15 | -43.90 | 12.45 | 0.79 |
| | 19970629 | 29-Jun | 30-Jul | 2171 | 1560 | 0.84 | 2470 | 13.77 | 9.82 | 7.26 | 0.83 | 2610 | 20.22 | 5.80 | 7.39 | 0.79 |
| | 19980630 | 30-Jun | 13-Jul | 4504 | 3050 | 0.78 | 3340 | -25.84 | 0.00 | -20.31 | 0.87 | 3010 | -33.17 | 1.64 | -20.24 | 0.87 |
| | 19980725 | 25-Jul | 2-Sep | 3698 | 3310 | 0.77 | 4030 | 8.98 | -1.28 | -13.54 | 0.83 | 3960 | 7.08 | -1.74 | -13.49 | 0.83 |
| | 20020622 | 22-Jun | 11-Jul | 5715 | 4560 | 0.81 | 6600 | 15.49 | -14.29 | 14.76 | 0.90 | 6390 | 11.81 | -22.45 | 14.78 | 0.86 |
| | 20020722 | 22-Jul | 4-Aug | 4290 | 2950 | 0.73 | 3150 | -26.57 | 46.94 | -30.09 | 0.83 | 3190 | -25.64 | 42.86 | -30.06 | 0.81 |
| | 20030622 | 22-Jun | 29-Jul | 8740 | 5030 | 0.66 | 7260 | -16.93 | 0.75 | -19.43 | 0.84 | 7140 | -18.31 | -2.99 | -19.44 | 0.80 |
| Validation | 20040717 | 17-Jul | 29-Jul | 2229 | 2020 | 0.44 | 1850 | -17.00 | -14.29 | 5.71 | 0.93 | 1810 | -18.80 | -18.37 | 5.88 | 0.87 |
| | 20040804 | 4-Aug | 13-Aug | 2641 | 1900 | 0.64 | 2870 | 8.67 | -12.24 | 9.79 | 0.95 | 2800 | 6.02 | -16.33 | 11.45 | 0.91 |
| | 20050707 | 7-Jul | 12-Aug | 7331 | 4070 | 0.65 | 8280 | 12.95 | -17.11 | -2.87 | 0.82 | 8350 | 13.90 | -25.00 | -3.08 | 0.63 |
| | 20050822 | 22-Aug | 10-Sep | 5650 | 3200 | 0.50 | 4320 | -23.54 | 1.65 | -18.78 | 0.71 | 4380 | -22.48 | -7.44 | -18.48 | 0.68 |

| | | | | | | | | | | | | | | | |
|---|---|---|---|---|---|---|---|---|---|---|---|---|---|---|---|
| 20060722 | 22-Jul | 16-Aug | 1770 | 1160 | 0.81 | 1580 | -10.73 | 1.82 | -26.86 | 0.92 | 1460 | -17.51 | -8.18 | -26.87 | 0.93 |
| 20070701 | 1-Jul | 1-Aug | 7926 | 5580 | 0.77 | 7980 | 0.68 | -6.50 | -16.70 | 0.83 | 7910 | -0.20 | -9.76 | -16.69 | 0.77 |
| 20080722 | 22-Jul | 9-Aug | 4264 | 3020 | 0.72 | 4460 | 4.60 | -4.08 | -7.25 | 0.89 | 4120 | -3.38 | -6.12 | -7.25 | 0.86 |
| 20080814 | 14-Aug | 27-Aug | 4219 | 2580 | 0.69 | 3440 | -18.46 | -12.90 | -5.11 | 0.82 | 3670 | -13.01 | -20.97 | -5.04 | 0.78 |
| 20090826 | 26-Aug | 13-Sep | 2221 | 1710 | 0.75 | 2710 | 22.02 | -4.17 | 7.64 | 0.89 | 2310 | 24.01 | -15.28 | 7.61 | 0.88 |
| 20100712 | 12-Jul | 5-Aug | 4314 | 3180 | 0.91 | 4370 | 1.30 | -2.63 | -14.68 | 0.92 | 4610 | 6.86 | -6.14 | -14.65 | 0.92 |
| Qualified (%) | | | | | | | 75 | 95.8 | 91.6 | | | 70.8 | 79.1 | 91.6 | |

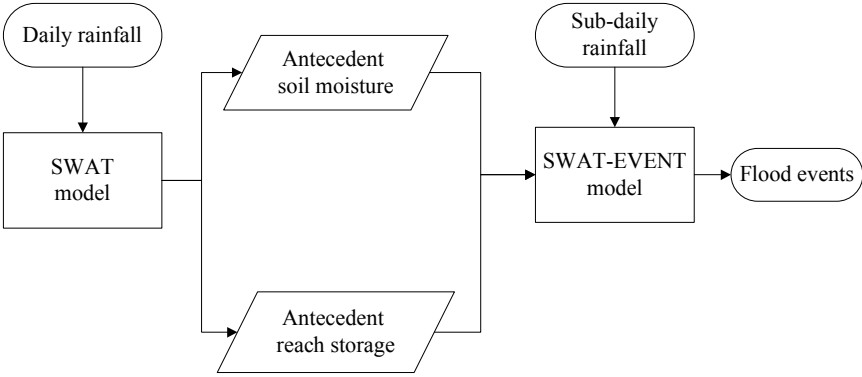

**Figure 1 The Wangjiaba (WJB) catchment.**

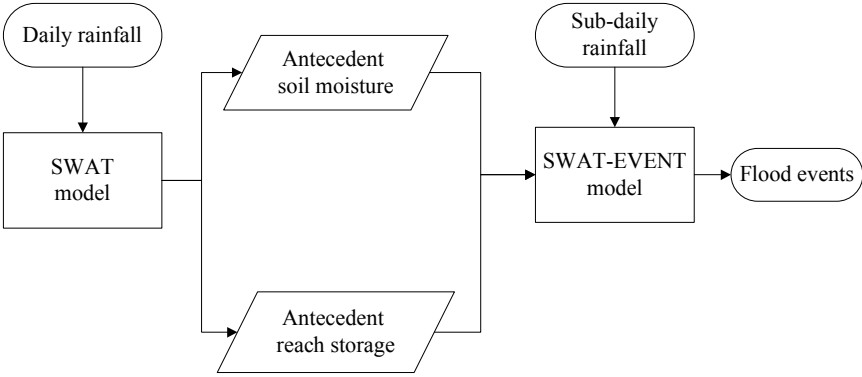

5    **Figure 2 SWAT-EVENT model for the simulation of event-based flood data based on the initial conditions extracted from daily simulation results produced by the original SWAT model.**

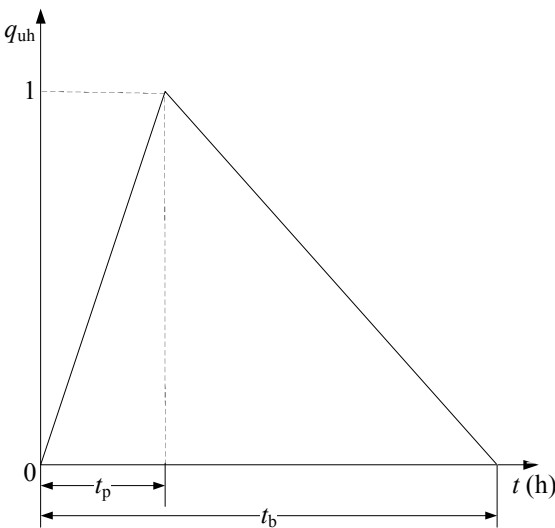

**Figure 3 Shape of the dimensionless triangular UH.**

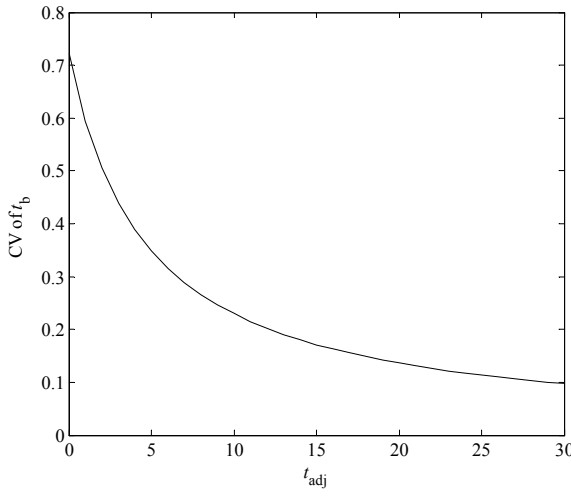

5    **Figure 4 Effect of basin level UH parameter $t_{adj}$ on the CV of UH time base $t_b$.**

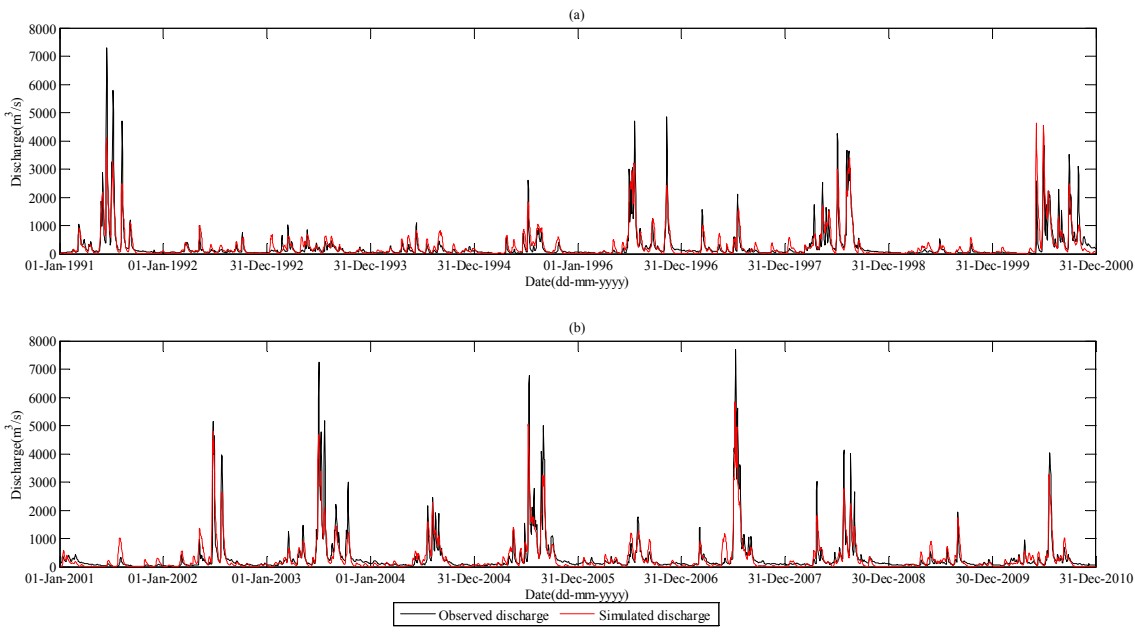

**Figure 5 Comparisons between the observed and simulated daily discharges for calibration (a) and validation (b) periods at WJB.**

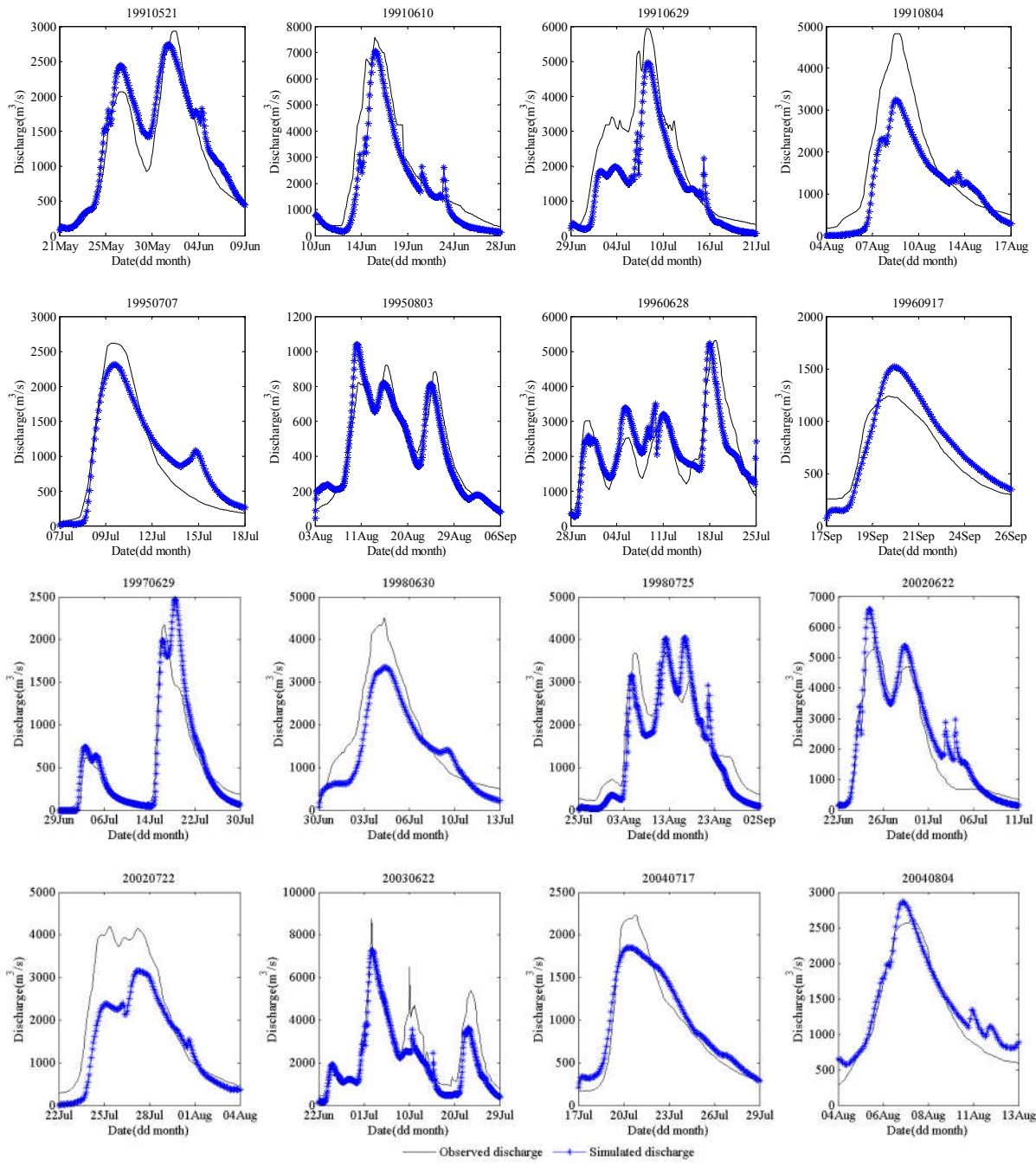

**Figure 6 Comparisons between the observed and simulated sub-daily flood events for the calibration period at WJB.**

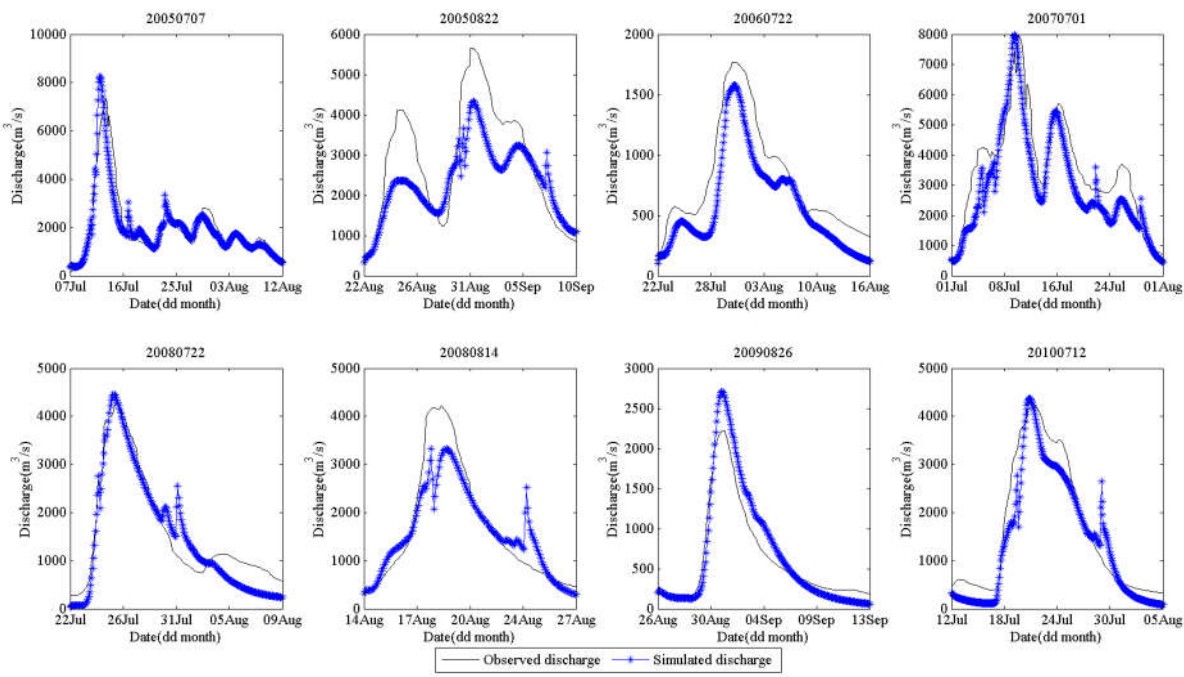

**Figure 7 Comparisons between the observed and simulated sub-daily flood events for the validation period at WJB**

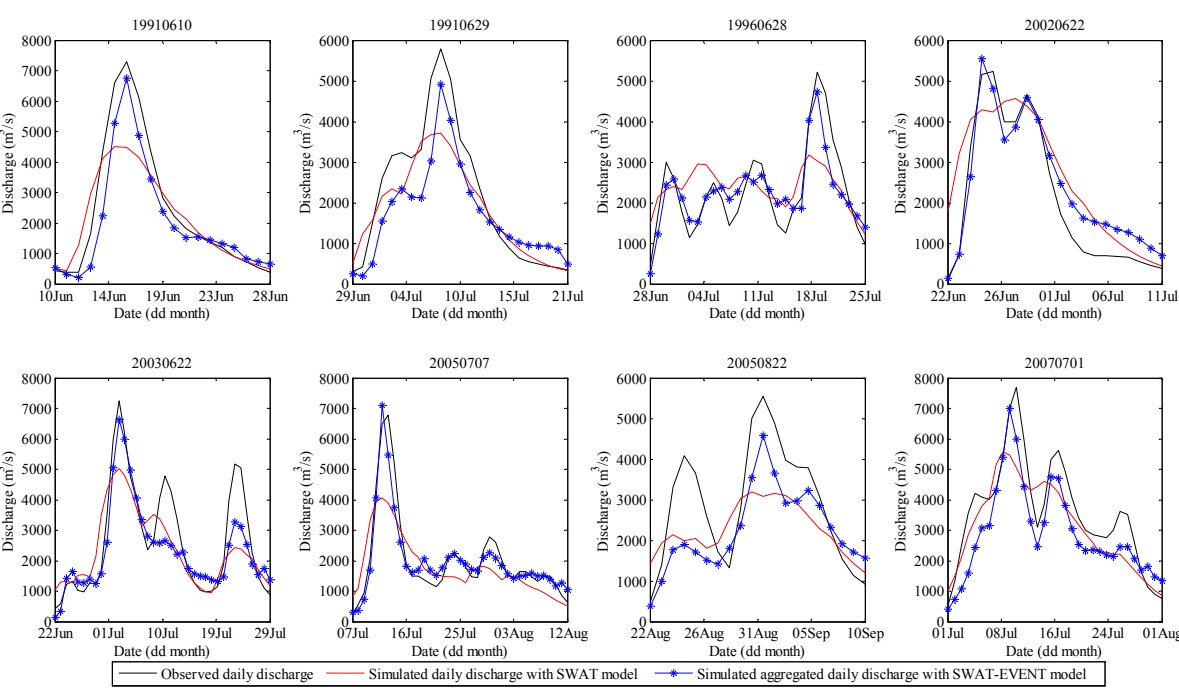

**Figure 8 Comparisons of the daily simulations conducted using the SWAT model and the aggregated sub-daily simulations conducted using the SWAT-EVENT model.**

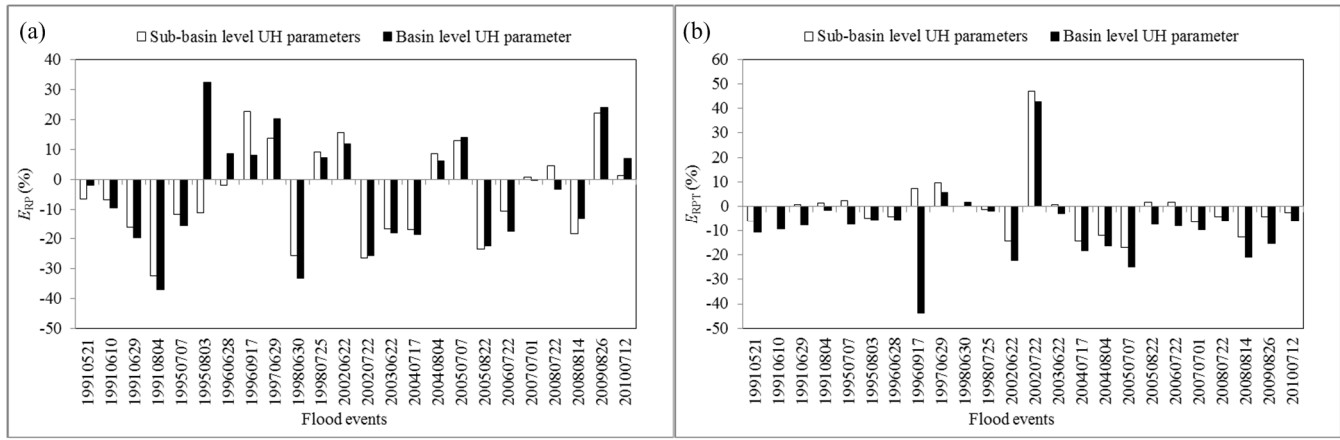

**Figure 9 Comparisons between sub-basin level and basin level UH parameter cases for relative peak discharge error (a) and relative peak time error (b).**

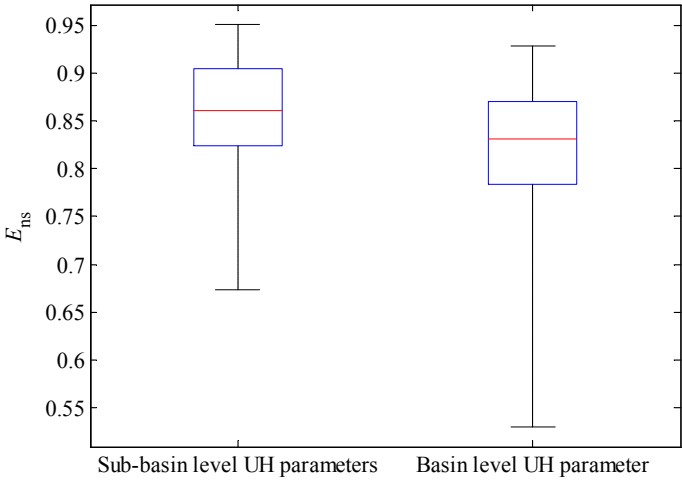

**Figure 10 Box plots of ENS values for the SWAT-EVENT model results for sub-basin level UH parameters and basin level UH parameters.**