# Peer review of "Improvement of the SWAT model for event-based flood simulation on a sub-daily time scale"

_Hydrology and Earth System Sciences, 2017_

## Referee Comment (RC1) · Anonymous Referee #1 · 21 Sep 2017

Review of the manuscript "Improvement of the SWAT model for event-based flood forecsting on a sub-daily time scale by Yu et al.

In this manuscript, Yu et al. suggest an improvement for the SWAT model version 2005 in two ways. First, spatial heterogeneity is inserted into the unit hydrograph parametrisation by using the SWAT model parameter on the subbasin level instead on the catchment level. Second, a sub-daily event-based modelling is suggested for the modelling of flood events. I see some interesting ideas in this study. However, I think that the manuscript needs to be strongly improved to identify its major strength in a better way. Thus, I recommend major revision.

Major comments:

The authors have to be more specific concerning the spatial resolution of the SWAT model. First, the SWAT model is a semi-distributed model consisting of subbasins and hydrological response units (HRU). Thus, please change from "distributed model" to "semi-distributed model".

Since the manuscript is focused on spatial heterogeneity, a clear description of the three levels in the SWAT model, namely catchment, subbasin and HRU level, is required.

Moreover, concerning the parameter heterogeneity and variability, it has to be clarified that some parameters (e.g. included in .bsn) are fixed for the whole catchment, others can be modified for each subbasins and a third group can be varied for each HRU. The authors should mention that the unit hydrograph is parameterized in the SWAT model version on the catchment level, which means that no spatial variation within a catchment is possible. In general, the SWAT model also allows a parameter variation on subbasin or HRU level. This has to be considered in the whole manuscript. In addition, the sentence in the abstract in L. 18 has to be more precise to avoid a misunderstanding.

The SWAT model version 2005 is a very old one. SWAT2009 and SWAT2012 are available since several years. Thus, please justify the use of SWAT2005. This is in particular relevant since the SWAT model was continuously improved and bugs were removed. Thus, the newer version are certainly better. Please give a statement on this.

The justification for the use of SWAT2005 (P.3, L. 13-15) cannot be accepted. Certainly it is possible to use SWAT-CUP for calibration, but it is certainly possible to use different calibration method for all SWAT version. There are lots of study using a different calibration approach for the SWAT model. Moreover a link between the use of SWAT2005 and the selected study catchment is not clear. Thus, please remove this part and provide a better explanation why SWAT2005 was selected instead of SWAT2009 or SWAT2012.

It is not acceptable to use only four sub-basins for 30630 km$^2$ catchment and claim at the same time limitations in spatial heterogeneity. According to my experience at least more than 100 subbasins can be expected for this catchment size. In particular, since the manuscript is focused on spatial heterogeneity, it is surprising that the subbasin number is very low.

Moreover, the SWAT model only provides spatially located outputs for each subbasin. In contrast, the authors stated that there are 138 gauges available. Thus, why do you not define a separate subbasin for each gauge or at least for the majority of the gauges? This would be even more a good approach to consider spatial heterogeneity.

The evaluation of the model results with Nash-Sutcliffe efficiency coefficient (ENS) and coefficient of determination (R2) is redundant. Both indices are mathematically closely related. R2 did not provide any additional knowledge about process or parameter behaviour. Even though that I am aware that there are still publications using ENS and R2, it is not anymore state-of-the-art. At the same time, the use of three performance criteria is recommended. Thus, please select in addition to ENS and PBIAS, a contrasting performance criteria which provides additional information and replace R2.

I do not think that your approach really shows an example for flood forecasting. It is a sub-daily model studies, but I do not see that there is a forecasting. The model is calibrated and validated. Could you please provide more information on how your approach is related to flood forecasting? Or say that this approach might be also beneficial for flood forecasting, but this was not considered here or will be part of future projects?

I have one general major comment: The authors suggest to improve the SWAT model in two ways. At first, at the spatial level, the model parameter t_adj is moved from the catchment level to the subbasin level to allow an individual parameterizsation for each subbasin and thus to consider spatial heterogeneity. Secondly, at the temporal scale, a sub-daily modeling is suggested to improve the representation of flood peaks.

Both aspects, spatial and temporal improvements, are not clearly enough separated. It would interesting to know why aspect improves the model more and in which part of the hydrograph. The study would benefit from a four-step comparison instead of a two-step comparison. To be more precise: I recommend to add two cases: (1) Sub-daily calculation with t_adj at catchment level (without t_subadj) and the opposite case (2) daily calculation with t_adj at subbasin level (with t_subadj).

The model modification of Jeong et al. (2010) to simulate on sub-daily resolution needs to be explained and not only mentioned (P.5, L.4). This is a core point of the manuscript. The readers need to understand this modification without reading the paper of Jeong et al. (2010).

You have mentioned that the SWAT is in its default version not adapted to sub-daily flood peak simulations. Keeping in mind that a large number of models is available: Why do you have selected the SWAT model and not a model which is focused on the hydrograph simulation. The major points of the SWAT model such as nutrient simulation, detailed land managements operation etc. are not relevant for your study.

The model results from SWAT2005 are used as input for SWAT-EVENT to simulation the flood peak. Is the model output of SWAT-EVENT then transfered back to SWAT2005? This point might be relevant since the first two flood events occured with a time lag of 19 days (P.9, L. 11). Thus, I expect a difference in the model states at the end of the first flood between SWAT and SWAT-EVENT. In this context, I like to mention that SWAT-EVENT does not impact the amount of water available in the system, but the water redistribution.

P.5, L.16-29: This part needs to be reformulated to present the idea in a better way. In the current version, it is difficult to understand.

P. 7, L. 1-2: This is a major point of the manuscript and has to be emphasised. A new model parameter is introduced at the subbasin level to include spatial heterogeneity. This is really important that it becomes clear.
P. 11, L. 18-19: This statement is not right. The SWAT model is not limited in representing low flows. It is more that there is a trade-off between high and low flow simulations at the same time. At it is true that it is difficult to represent high and low flows in a very good quality with the same model run. This is by the way an often occurring problem in hydrological modelling. The major point here is that the selection of the performance measures is at the same time a decision on the study focus. By selecting the Nash-Sutcliffe Efficiency high flows are more weighted than low flows. Thus, it would not be a surprise if the high flows are well represented while low flows perform poorly. This results could be different if using ENSlog or a different performance measure related to low flows. Please improve this statement.

The aspect of flood forecasting is strongly emphasised in the conclusion. I still do not see that the strength of the manuscript is related to flood forecasting. Please rework the conclusion accordingly.

Specific comments:

P.3, L.3: How do you "relate hydrologic response to specific catchment characteristics"? By parameter settings?

P.4, L. 18: The weather generator is only used in the case of missing climate data. Please improve this statement.

P. 5, L. 21: It was not mentioned before that the SWAT model includes HRUs. Please improve the description of the SWAT model.

P. 7, L. 9: Please denote the 26 parameters, maybe in the attachments.

P. 8, L.13: Please explain the three indices, at best with equations.

---

## Referee Comment (RC2) · Anonymous Referee #2 · 22 Sep 2017

General Comment: I would expect that in 2017 SWAT modeller would use the newest version of SWAT 2012 especially as next year SWAT+ a new generation of the model will be presented. However I can understand that simpler structure of the 2005 version is easily manageable and modified when you start with this kind of research.

Introduction P3, L13-14: Please better justify selection of SWAT 2005. Current justification is not satisfactory. 2.2 Model dataset P4, L18-21: I am surprised that you used Weather generator. That is really rare. The area is very large I would expect to have at least some data. Did you also use it for precipitation? And why data back to 1979 if you're modelling period 1991 – 2010. How did you model land use management (.mgt) where did you obtain the data. Please add table with data used in the model. For example refer to this manuscripts: Glavan, M., Ceglar, A. and Pintar, M., 2015. Assessing

the impacts of climate change on water quantity and quality modelling in small Slovenian Mediterranean catchment - lesson for policy and decision makers. Hydrological Processes, 29(14): 3124-3144.

3.1 Development of... P5, L4: If you are following the method proposed by Jeong et al. (2010) please describe why and for what purpose was it made or used.

3.3 Model calibration Please introduce table with parameters used in calibration. Include also default value, range, final value. For example refer to this manuscripts: Glavan, M., Ceglar, A. and Pintar, M., 2015. Assessing the impacts of climate change on water quantity and quality modelling in small Slovenian Mediterranean catchment - lesson for policy and decision makers. Hydrological Processes, 29(14): 3124-3144. This manuscript should also be part of introduction or discussion chapters as it clearly describes the process that need to be followed while using SWAT model. Please clearly describe what the scenarios were. I assume you had three scenarios as follows out from Table 3 where you presented for certain version (I assume SWAT-EVENT, please write this in title of the table) three scenarios Daily simulation, Basin level UH parameter simulation and Sub-basin level UH parameter simulation. From Figure & I can see you had two scenarios Simulated daily discharge SWAT and simulated sub-daily discharge SWAT-EVENT. In methodologies clearly describe what is base scenario and to which scenario is it compared.

5 Discussion P10, L28-30: Sentences from previous chapters are often repeated.

6. Conclusions P12, L16-30: All the text in the conclusions is just repeated from previous chapters. Delete existent text and please write down answers to this questions in conclusions: Why is this research unique? What are the shortcomings/uncertainties of this research? What did us and science community learned from it? Future work?

---

## Author Comment (AC1) · 17 Nov 2017

**Response to Interactive comment Anonymous Referee #1**

We heartily appreciate the reviewer's assessment on this study and the valuable suggestions provided to improve this manuscript. We hereby provide our point by point responses how the comments by referee #1 will be addressed in the revised manuscript.

**1. Responses to major comments**

*Comment: The authors have to be more specific concerning the spatial resolution of the SWAT model. First, the SWAT model is a semi-distributed model consisting of subbasins and hydrological response units (HRU). Thus, please change from "distributed model" to "semi-distributed model".*

Reply: Thanks for your correction. The "distributed hydrological model" is changed to "semi-distributed hydrological model" in the manuscript.

*Comment: Since the manuscript is focused on spatial heterogeneity, a clear description of the three levels in the SWAT model, namely catchment, subbasin and HRU level, is required.*

Reply: We appreciate the comment. It is really necessary to state the spatial discretization of the SWAT model. The following statement is added to the third paragraph in section 1:

"To spatially characterize the inhomogeneity, the SWAT model delineates a catchment into a number of sub-basins, which were subsequently divided into Hydrologic Response Units (HRUs). In SWAT model, HURs are basic simulation units of the land phase of the hydrological cycle that controls the total yield of streamflow, sediment, pesticide and nutrient to the main channel in corresponding sub-basin. Afterwards the routing phase converges the land phase results to the watershed outlet through the channel network."

*Comment: Moreover, concerning the parameter heterogeneity and variability, it has to be clarified that some parameters (e.g. included in .bsn) are fixed for the whole catchment, others can be modified for each subbasins and a third group can be varied for each HRU. The authors should mention that the unit hydrograph is parameterized in the SWAT model version on the catchment level, which means that no spatial variation within a catchment is possible. In general, the SWAT model also allows a parameter variation on subbasin or HRU level. This has to be considered in the whole manuscript. In addition, the sentence in the abstract in L. 18 has to be more precise to avoid a misunderstanding.*

Reply: Thank you very much for your advice. We've realized that the parameter heterogeneity and variability are important issues in distributed or semi-distributed hydrological model application.

The following description is added to the fifth paragraph in section 1:

"Due to the spatial discretization in the SWAT model, the model parameters are categorized into three levels: (1) basin level parameters are fixed for the whole catchment; (2) sub-basin level parameters are varied with sub-basins; (3) HRU level parameters are distributed in different HRUs."

The sentence "In addition, the SWAT model provides a uniform parameter set with which to adjust the shape of the UH in each sub-basin." in the fifth paragraph in section 1 is rephrased as follows:

"By default, the UH related parameters in the SWAT model are on the basin level, which indicates that no spatial variation within a catchment is possible when adjusting the shape of the UH in each sub-basin.".

The sentence in the abstract in L. 18 is changed as follows:

"SWAT uses a basin level parameter that is fixed for the whole catchment to parameterize the Unit Hydrograph (UH), thereby ignoring the spatial heterogeneity among the sub-basins when adjusting the shape of the UHs."

*Comment: The SWAT model version 2005 is a very old one. SWAT2009 and SWAT2012 are available since several years. Thus, please justify the use of SWAT2005. This is in particular relevant since the SWAT model was continuously improved and bugs were removed. Thus, the newer versions are certainly better. Please give a statement on this.*

*The justification for the use of SWAT2005 (P.3, L. 13-15) cannot be accepted. Certainly it is possible to use SWAT-CUP for calibration, but it is certainly possible to use different calibration method for all SWAT version. There are lots of study using a different calibration approach for the SWAT model. Moreover a link between the use of SWAT2005 and the selected study catchment is not clear. Thus, please remove this part and provide a better explanation why SWAT2005 was selected instead of SWAT2009 or SWAT2012.*

Reply: We propose to add the following for justification:

"The SWAT2005 version has an existing calibration module while the SWAT2009 and the SWAT2012 have removed the autocalibration routines. The integrated design of model simulation and autocalibration in the SWAT2005 is easily manageable and modified since there is no need to couple other algorithms. According to the revision history of the SWAT model, revisions after the SWAT2005 aimed mainly at the water quality simulation and have little effect on runoff simulation. Thus the SWAT2005 is employed in this study."

And the revision history will be provided as the attachments.

***Comment: It is not acceptable to use only four sub-basins for 30630 km² catchment and claim at the same time limitations in spatial heterogeneity. According to my experience at least more than 100 subbasins can be expected for this catchment size. In particular, since the manuscript is focused on spatial heterogeneity, it is surprising that the subbasin number is very low.***

Reply: We agree with the reviewer for this comment. Sub-basin is assumed homogeneous with parameters at the sub-basin level. Since this paper discussed the spatial differences in model parameter, we are going to redefine the sub-basins and do all the simulations again. We still need time to organize this part and will present the modification in revised manuscript.

***Comment: Moreover, the SWAT model only provides spatially located outputs for each subbasin. In contrast, the authors stated that there are 138 gauges available. Thus, why do you not define a separate subbasin for each gauge or at least for the majority of the gauges? This would be even more a good approach to consider spatial heterogeneity.***

Reply: Since we consider redefinition of the sub-basins, we will use the Taisen Polygon Method to calculate area rainfall in each sub-basin to consider the spatial heterogeneity of rainfall gages.

***Comment: The evaluation of the model results with Nash-Sutcliffe efficiency coefficient (ENS) and coefficient of determination (R2) is redundant. Both indices are mathematically closely related. R2 did not provide any additional knowledge about process or parameter behaviour. Even though that I am aware that there are still publications using ENS and R2, it is not anymore state-of-the-art. At the same time, the use of three performance criteria is recommended. Thus, please select in addition to ENS and PBIAS, a contrasting performance criteria which provides additional information and replace R2.***

Reply: We appreciate the comment. We would like to use the root mean square error ( $R_{\mathrm{MSE}}$ ) instead of $R^2$. $R_{\mathrm{MSE}}$ indicates a perfect match between observed and predicted values when it equals 0, with increasing $R_{\mathrm{MSE}}$ values indicating an increasingly poor match. The related descriptions will be modified in the revised manuscript.

***Comment: I do not think that your approach really shows an example for flood forecasting. It is a sub-daily model studies, but I do not see that there is a forecasting. The model is calibrated and validated. Could you please provide more information on how your approach is related to flood forecasting? Or say***

*that this approach might be also beneficial for flood forecasting, but this was not considered here or will be part of future projects?*

Reply: Thanks for your comment. We want to make some explanations here.

The hydrological model itself is not an example for flood forecasting, but a core part of the flood forecasting system. The quality of the hydrological model (in terms of its structure and parameter estimates) is one of the important factors for accurate flood forecasting (Noh et al., 2014). Thus this paper mainly focused on the modification of the structure of the original SWAT model to verify its suitability for flood simulation in study area, which is tightly related to the flood forecasting.

Meantime, model parameter estimation is an inevitable issue accompanied by the application of the hydrological model. Typically, calibration is performed by using multiple historical flood events data. Subsequently, the model validation consists in running the model under another group of historical flood events, with the input of parameter values thus being estimated in the calibration phase. This kind of calibration and validation is the common solution in many flood forecasting practices (Hapuarachchi and Wang, 2008). Thus the modified SWAT model was calibrated with 16 flood events from 1991 to 2004 and validated with 8 flood events from 2005 to 2010 in this paper.

In addition, Berthet et al. (2009) declared that the major drawback of the continuous simulation lies in its data requirements: long continuous precipitation time series up to the day of interest are difficult to obtain in an operational forecasting perspective. Yao et al. (2014) claimed that long continuous daily simulations are implemented to compute the soil moisture states that are used as antecedent conditions for the flood events in the operational use. Therefore the Figure 3 in this paper showed the operation at two time scales (i.e., continuous daily simulation and event-base sub-daily simulation).

In summary, this study addressed the model-based problems that related to flood forecasting. However we think it is still necessary to further clarify the relationship between the flood forecasting and our approaches.

The second paragraph in section 1: "A large number of distributed or semi-distributed hydrological models have been applied in flood forecasting (BEVEN and KIRKBY, 1979;Singh, 1997;Xiong and Guo, 2004;Mendes and Maia, 2017;Hapuarachchi et al., 2011)." replaces the "The Soil and Water Assessment Tool (SWAT) model (Arnold et al., 1998;Srinivasan et al., 1998;Arnold and Fohrer, 2005) is the most widely used of the prevailing distributed models." to emphasize the model-based approaches of flood forecasting.

We propose a paragraph explaining the approaches that have addressed the model structures and parameters issues in flood forecasting in the conclusions section:

"Flood forecasting is a synthetic system that integrates the data acquisition and processing, rainfall-runoff modeling and warning information release etc. Hydrological models are always the core part of the forecasting system. Model structures and model parameters are one of the most important issues for accurate flood forecasting (Noh et al., 2014). The original SWAT model is not competent to flood forecasting due to its initial design of long-term simulations with daily time-steps. This paper mainly focused on the modification of the structure of the original SWAT model to perform event-based simulation, which was applicable for the area without continuous long-term observations. The newly developed SWA-EVENT model was applied in the upper reaches of the Huaihe River. Model calibration and validation were made by the using of historical flood events, showing good simulation accuracy. To improve the spatial representation of the SWA-EVENT, the lumped UH parameters were then adjusted to the distributed ones. Calibration and validation results revealed the improvement of event-based simulation performances. This study expands the application of the original SWAT model in event-based flood simulation."

*Comment: I have one general major comment: The authors suggest to improve the SWAT model in two ways. At first, at the spatial level, the model parameter t_adj is moved from the catchment level to the subbasin level to allow an individual parameterizsation for each subbasin and thus to consider spatial heterogeneity. Secondly, at the temporal scale, a sub-daily modeling is suggested to improve the representation of flood peaks. Both aspects, spatial and temporal improvements, are not clearly enough separated. It would interesting to know why aspect improves the model more and in which part of the hydrograph. The study would benefit from a four-step comparison instead of a two-step comparison. To be more precise: I recommend to add two cases: (1) Sub-daily calculation with t_adj at catchment level (without t_subadj) and the opposite case (2) daily calculation with t_adj at subbasin level (with t_subadj).*

Reply: Thanks for your comment. We assume the referee is here referring the four-step comparison:

daily calculation with catchment level parameter (t_adj);

daily calculation with sub-basin level parameter (t_subdadj);

sub-daily calculation with catchment level parameter (t_adj);

sub-daily calculation with sub-basin level parameter (t_subdadj).

If so, we want to respectfully clarify some confusions. In SWAT model, the Unit Hydrograph (UH)

method is only used for sub-daily simulation, rather than a daily simulation. The main reason of this situation is that the flood hydrograph resulting from a known storm often vary significantly within sub-daily time-scales, while the daily calculation may exceed the time of concentration for most of the sub-basins. Even for large sub-basins with a time of concentration greater than 1 day, SWAT incorporates a lag equation to store a portion of the surface runoff release to the main channel. Thus we think there are no such cases: daily calculation with catchment level parameter (t_adj) and daily calculation with sub-basin level parameter (t_subdadj).

The other argument is that modification at the temporal level was the first step, and modification at the spatial level was the second step.

At the temporal level, there are two drawbacks in the application of the original SWAT model for event-based flood simulation: (1) algorithms with daily time step for some hydrological processes are implicitly assumed (2) the continuous long-term simulation loop of its initial design. This paper referenced the methodologies in a previous study (Jeong et al., 2010) to solve the aforementioned problem (1). Then this paper broke down the continuous cycle of the model structure to solve the problem (2). Finally the SWAT-EVENT model was developed to simulate the event-based floods. At the spatial level, the UH parameter was modified from basin level to sub-basin level to represent the spatial heterogeneity of studied catchment, expecting an improvement for event-based flood simulation with the SWAT-EVENT model.

This paper used a two-step comparison to prove that: (1) temporal modification enabled the original SWAT model to simulate flood events and the improvements of the aggregated daily performances of the SWAT-EVENT model in Figure 6 were due to the higher temporal resolutions for input rainfall and the simulation time step; (2) spatial modification improved the simulation accuracy for even-based floods (Table 3 and Figure 8).

*Comment: The model modification of Jeong et al. (2010) to simulate on sub-daily resolution needs to be explained and not only mentioned (P.5, L.4). This is a core point of the manuscript. The readers need to understand this modification without reading the paper of Jeong et al. (2010).*

Reply: Thank you for your valuable suggestion. We will explain specifically the model modification of Jeong et al. (2010) in the revised manuscript. I still need time to organize this part.

*Comment: You have mentioned that the SWAT is in its default version not adapted to sub-daily flood peak simulations. Keeping in mind that a large number of models is available: Why do you have selected*

*the SWAT model and not a model which is focused on the hydrograph simulation. The major points of the SWAT model such as nutrient simulation, detailed land managements operation etc. are not relevant for your study.*

Reply: The perceptive comment shows the reviewer's knowledge in the field, and we have to admit that the SWAT model is not the first choice of flood simulation because there are so many other models have good applicability in this field. Here we respectfully argue that this study still has certain scientific significance. Though the highlights of the model of the SWAT model are the predictions of the impact of land management practices on water, sediment and agricultural chemical yields, runoff simulation is always a fundamental. Moreover, we have noticed that the study on the event-based water quality assessment has been a hot topic (He et al., 2010;Nguyen and Meon, 2013;He et al., 2011). Therefore the improvement of the SWAT model for event-based flood simulation will lay the lay the foundation for event-based water quality modeling. To emphasize our points, the following statement is added to the conclusion section:

"Event-based runoff quantity and quality modeling has become a challenge task since the impact of hydrological extremes on the water quality is particularly important. The improvement of the SWAT model for event-based flood simulation will lay the foundation for dealing with the event-based water quality issues."

*Comment: The model results from SWAT2005 are used as input for SWAT-EVENT to simulation the flood peak. Is the model output of SWAT-EVENT then transfered back to SWAT2005? This point might be relevant since the first two flood events occured with a time lag of 19 days (P.9, L. 11). Thus, I expect a difference in the model states at the end of the first flood between SWAT and SWAT-EVENT. In this context, I like to mention that SWAT-EVENT does not impact the amount of water available in the system, but the water redistribution.*

Reply: The answer to the question "Is the model output of SWAT-EVENT then transfered back to SWAT2005" is no. In fact, the daily SWAT model and the sub-daily SWAT-EVENT model were executed independently. According to Figure 3, the continuous daily SWAT model ran first to calculate the antecedent conditions for each flood events. After this, the SWAT-EVENT model began to run. This approach has been commonly used in other researches when no measurements of antecedent moisture conditions are available (Tramblay et al., 2012).

*Comment: P.5, L.16-29: This part needs to be reformulated to present the idea in a better way. In the*

*current version, it is difficult to understand.*

Reply: We agree that better reformulation about the modification of the SWAT model should be presented. Reorganization of this part will be completed in the revised manuscript. We still need time to organize this part.

*Comment: P. 7, L. 1-2: This is a major point of the manuscript and has to be emphasised. A new model parameter is introduced at the subbasin level to include spatial heterogeneity. This is really important that it becomes clear.*

Reply: Thank you for the suggestion. We think it is really necessary to add more descriptions for this major point. The last paragraph in section 3.2 is changed to:

"According to Eq. (4) the time base of the UH ($t_b$) is determined by both concentration time for the sub-basin ($t_c$) and shape adjustment factor ($t_{adj}$) concurrently. Though the $t_c$ can present the spatial differences among sub-basins based on the geographical characters including slope length, slope steepness and sub-basin area and et al. The variable $t_{adj}$ in Eq. (4) is a basin level parameter possessing the same value for the whole catchment, meaning that the spatial heterogeneities may be homogenized. As seen in Table 1, the values of the average slope steepness ($S_{sub}$) of HC and XX are much higher than those of BT and WJB. Meanwhile, the average slope lengths ($L_{slp}$) for HC and XX are shorter than those for BT and WJB. Thus, to highlight the differences representative of the UHs between each of the sub-basins, the parameter $t_{adj}$ was modified from the basin level to the sub-basin level and renamed $t_{subadj}$."

*Comment: P. 11, L. 18-19: This statement is not right. The SWAT model is not limited in representing low flows. It is more that there is a trade-off between high and low flow simulations at the same time. At it is true that it is difficult to represent high and low flows in a very good quality with the same model run. This is by the way an often occurring problem in hydrological modelling. The major point here is that the selection of the performance measures is at the same time a decision on the study focus. By selecting the NashSutcliffe Efficiency high flows are more weighted than low flows. Thus, it would not be a surprise if the high flows are well represented while low flows perform poorly. This results could be different if using ENSlog or a different performance measure related to low flows. Please improve this statement.*

Reply: Thank you very much for pointing out my mistake. The following statement is going to correct

the mistake:

"On the one hand, the SWAT model was calibrated using the sum of squares of the residuals as the objective function, which was more sensitive to high flows than low flows. Thus the calibration results ensured the simulation accuracy at the expense of the low flow performances"

***Comment: The aspect of flood forecasting is strongly emphasised in the conclusion. I still do not see that the strength of the manuscript is related to flood forecasting. Please rework the conclusion accordingly.***

Reply: We will rework the whole conclusion section as follows:

"Flood forecasting is a synthetic system that integrates the data acquisition and processing, rainfall-runoff modeling and warning information release etc. Hydrological models are always the core part of the forecasting system. Model structures and model parameters are one of the most important issues for accurate flood forecasting (Noh et al., 2014). The original SWAT model was not competent to flood forecasting due to its initial design of long-term simulations with daily time-steps. This paper mainly focused on the modification of the structure of the original SWAT model to perform event-based simulation, which was applicable for the area without continuous long-term observations. The newly developed SWA-EVENT model was applied in the upper reaches of the Huaihe River. Model calibration and validation were made by the using of historical flood events, showing good simulation accuracy. To improve the spatial representation of the SWA-EVENT, the lumped UH parameters were then adjusted to the distributed ones. Calibration and validation results revealed the improvement of event-based simulation performances. This study expands the application of the original SWAT model in event-based flood simulation.

The determination of hydrological model parameters is an inevitable process before flood forecasting. Parameter estimations of distributed or semi-distributed hydrological models commonly depend on automated calibration procedure due to overparametrization. The optimal parameters of the SWAT-EVENT model were obtained by the automatic parameter calibration module that integrated SCE-UA algorithm in this study. However, serveral factors such as interactions among model parameters, complexities of spatio-temporal scales and statistical features of model residuals may lead to the parameter non-uniqueness, which is the source of the uncertainty in the estimated parameters. Uncertainty of model parameters will be finally passed to the model results, hence leading to certain risks in flood forecasting. In the future, emphasis will be placed on the quantification of the parameter uncertainty to provide better supports for flood

operations.

Event-based runoff quantity and quality modeling has become a challenge task since the impact of hydrological extremes on the water quality is particularly important. The improvement of the SWAT model for event-based flood simulation will lay the foundation for dealing with the event-based water quality issues."

**2. Responses to specific comments**

*Comment: P.3, L.3: How do you "relate hydrologic response to specific catchment characteristics"? By parameter settings?*

Reply: The dimensionless UH used in the SWAT model is just one form of the Synthetic Unit Hydrograph (SUH), which can be used to the ungauged catchments. The SUH was derived from catchment characteristics rather than rainfall-runoff data (Bhunya, 2011). According to Equation (3), the UH was defined based on the hydrologic property of the catchment. To be clear, detail descriptions of this kind of UH will be added in the revised manuscript.

*Comment: P.4, L. 18: The weather generator is only used in the case of missing climate data. Please improve this statement.*

Reply: We suggest the following statement to be a replacement:

"The SWAT model has developed a weather generator (WXGEN) to fill the missing climate data by the use of monthly statistics."

*Comment: P. 5, L. 21: It was not mentioned before that the SWAT model includes HRUs. Please improve the description of the SWAT model.*

Reply: We suggest the following description to be added to the third paragraph in section 1:

"To spatially characterize the inhomogeneity, the SWAT model delineates a catchment into a number of sub-basins, which were subsequently divided into Hydrologic Response Units (HRUs). In SWAT model, HURs are basic simulation units of the land phase of the hydrological cycle that controls the total yield of streamflow, sediment, pesticide and nutrient to the main channel in corresponding sub-basin. Afterwards the routing phase converges the land phase results to the watershed outlet through the channel network."

*Comment: P. 7, L. 9: Please denote the 26 parameters, maybe in the attachments.*

Reply: Thanks for the good suggestion. The following table is added to introduce the 26 parameters in the SWAT model:

| Parameters | Definition | lower bound | upper bound |
|---|---|---|---|
| ALPHA_BF | Baseflow alpha factor (days). | 0 | 1 |
| BIOMIX | Biological mixing efficiency. | 0 | 1 |
| BLAI | Maximum potential leaf area index. | 0 | 1 |
| CANMX | Maximum canopy storage (mm H2O). | 0 | 10 |
| CH_K(2) | Effective hydraulic conductivity in main channel alluvium (mm/hr). | 0 | 150 |
| CH_N | Manning's "n" value for the main channel. | 0 | 1 |
| CN2 | Initial SCS runoff curve number for moisture condition II. | -25 | 25 |
| EPCO | Plant uptake compensation factor. | 0 | 1 |
| ESCO | Soil evaporation compensation factor | 0 | 1 |
| GW_DELAY | Groundwater delay time (days). | -10 | 10 |
| GW_REVAP | Groundwater "revap" coefficient. | -0.036 | 0.036 |
| GWQMN | Threshold depth of water inthe shallow aquifer required for return flow to occur (mm H2O). | -1000 | 1000 |
| REVAPMN | Threshold depth of water in the shallow aquifer for "revap" or percolation to the deep aquifer to occur (mm H2O). | -100 | 100 |
| SMTMP | Snow melt base temperature (ºC). | 0 | 5 |
| SLOPE | Average slope | -25 | 25 |
| SLSUBBSN | Average slope length (m). | -25 | 25 |
| SMFMN | Melt factor for snow on December 21 (mm H2O/ºC-day). | 0 | 10 |
| SMFMX | Melt factor for snow on June 21 (mm H2O/ºC-day). | 0 | 10 |
| SMTMP | Snow melt base temperature (ºC). | -25 | 25 |
| SOL_ALB | Moist soil albedo. | -25 | 25 |
| SOL_AWC | Available water capacity of the soil layer (mm H2O/mm soil). | -25 | 25 |
| SOL_K | Saturated hydraulic conductivity (mm/hr). | -25 | 25 |
| SOL_Z | Depth from soil surface to bottom of layer (mm). | -25 | 25 |
| SURLAG | Surface runoff lag coefficient. | 0 | 10 |
| TIMP | Snow pack temperature lag factor. | 0 | 1 |
| TLAPS | Temperature lapse rate (ºC/km). | 0 | 50 |

***Comment: P. 8, L.13: Please explain the three indices, at best with equations.***

Reply: We agree to the comment and will add the following equations for the three indices in the revised manuscript:

$$E_{NS} = 1 - \left[ \frac{\sum_{i=1}^{n} \left( Q_{obs}(i) - Q_{sim}(i) \right)^2}{\sum_{i=1}^{n} \left( Q_{obs}(i) - \overline{Q}_{obs} \right)^2} \right]$$

$$P_{BIAS} = \left[ \frac{\sum_{i=1}^{n} \left( Q_{obs}(i) - Q_{sim}(i) \right) \cdot 100}{\sum_{i=1}^{n} Q_{obs}(i)} \right]$$

$$R_{\mathrm{MSE}} = \sqrt{\frac{\sum_{i=1}^{n}\left(Q_{obs}(i) - Q_{sim}(i)\right)^2}{n}}$$

where $Q_{obs}$ is the observed values; $Q_{sim}$ is the simulated values; $n$ is the number of the value points.

**References**

Arnold, J. G., Srinivasan, R., Muttiah, R. S., and Williams, J. R.: LARGE AREA HYDROLOGIC MODELING AND ASSESSMENT PART I: MODEL DEVELOPMENT 1, JAWRA Journal of the American Water Resources Association, 34, 73–89, 1998.

Arnold, J. G., and Fohrer, N.: SWAT2000: Current Capabilities and Research Opportunities in Applied Watershed Modeling, Hydrol. Process., 19, 563-572, 2005.

Berthet, L., Andréassian, V., Perrin, C., and Javelle, P.: How crucial is it to account for the antecedent moisture conditions in flood forecasting? Comparison of event-based and continuous approaches on 178 catchments, Hydrology & Earth System Sciences, 13, 819-831, 2009.

BEVEN, K. J., and KIRKBY, M. J.: A physically based, variable contributing area model of basin hydrology / Un modèle à base physique de zone d'appel variable de l'hydrologie du bassin versant, Hydrological Sciences Bulletin, 24, 43-69, 1979.

Bhunya, P. K.: Synthetic Unit Hydrograph Methods: A Critical Review, Open Hydrology Journal, 5, 1-8, 2011.

Hapuarachchi, H. A. P., and Wang, Q. J.: A review of methods and systems available for flash flood forecasting, 2008.

Hapuarachchi, H. A. P., Wang, Q. J., and Pagano, T. C.: A review of advances in flash flood forecasting, Hydrol. Process., 25, 2771-2784, 2011.

He, J., Valeo, C., Chu, A., and Neumann, N. F.: Characteristics of suspended solids, microorganisms, and chemical water quality in event-based stormwater runoff from an urban residential area, Water Environment Research A Research Publication of the Water Environment Federation, 82, 2333, 2010.

He, J., Valeo, C., Chu, A., and Neumann, N. F.: Prediction of event-based stormwater runoff quantity and quality by ANNs developed using PMI-based input selection, J. Hydrol., 400, 10-23, 2011.

Jeong, J., Kannan, N., Arnold, J., Glick, R., Gosselink, L., and Srinivasan, R.: Development and Integration of Sub-hourly RainfallRunoff Modeling Capability Within a Watershed Model, General Information, 24, 4505-4527, 2010.

Mendes, J., and Maia, R.: Hydrologic Modelling Calibration for Operational Flood Forecasting, Water Resour. Manag., 30, 1-15, 2017.

Nguyen, H. Q., and Meon, G.: SINUDYM - an event-based water quality model for ungauged catchments, EGU General Assembly Conference, 2013,

Noh, S. J., Rakovec, O., Weerts, A. H., and Tachikawa, Y.: On noise specification in data assimilation schemes for improved flood forecasting using distributed hydrological models, J. Hydrol., 519, 2707-2721, 2014.

Singh, V. P.: Computer models of watershed hydrology, Computer Models of Watershed Hydrology, 443-476, 1997.

Srinivasan, R., Ramanarayanan, T. S., Arnold, J. G., and Bednarz, S. T.: Large Area Hydrologic Modeling and Assessment: Part II – Model Application, JAWRA Journal of the American Water Resources Association, 34, 91-101, 1998.

Tramblay, Y., Bouaicha, R., Brocca, L., Dorigo, W., Bouvier, C., Camici, S., and Servat, E.: Estimation of antecedent wetness conditions for flood modelling in northern Morocco, Hydrology & Earth System Sciences Discussions, 9, 4375-4386, 2012.

Xiong, L., and Guo, S.: Effects of the catchment runoff coefficient on the performance of TOPMODEL in rainfall–runoff modelling, Hydrol. Process., 18, 1823–1836, 2004.

Yao, C., Zhang, K., Yu, Z., Li, Z., and Li, Q.: Improving the flood prediction capability of the Xinanjiang model in ungauged nested catchments by coupling it with the geomorphologic instantaneous unit hydrograph, J. Hydrol., 517, 1035-1048, http://dx.doi.org/10.1016/j.jhydrol.2014.06.037, 2014.

---

## Author Comment (AC2) · 17 Nov 2017

**Response to Interactive comment Anonymous Referee #2**

We heartily appreciate the reviewer's assessment on this study and the valuable suggestions provided to improve this manuscript. We hereby provide our point by point responses how the comments by referee #2 will be addressed in the revised manuscript.

*Comment: General Comment: I would expect that in 2017 SWAT modeller would use the newest version of SWAT 2012 especially as next year SWAT+ a new generation of the model will be presented. However I can understand that simpler structure of the 2005 version is easily manageable and modified when you start with this kind of research. Introduction P3, L13-14: Please better justify selection of SWAT 2005. Current justification is not satisfactory.*

Reply: We propose to add the following for justification:

"The SWAT2005 version has an existing calibration module while the SWAT2009 and the SWAT2012 have removed the autocalibration routines. The integrated design of model simulation and autocalibration in the SWAT2005 is easily manageable and modified since there is no need to couple other algorithms. According to the revision history of the SWAT model, revisions after the SWAT2005 aims mainly at the water quality simulation and has little effect on runoff simulation. Thus the SWAT2005 is employed in this study."

And the revision history will be provided as the attachments.

*Comment: 2.2 Model dataset P4, L18-21: I am surprised that you used Weather generator. That is really rare. The area is very large I would expect to have at least some data. Did you also use it for precipitation? And why data back to 1979 if you're modelling period 1991 – 2010.*

Reply: Thanks for the comments. We want to make some explanations here. Weather generator is only used in the case of missing climate data. In this study only the observed rainfall data were available while the other climatic data such as relative humidity, wind speed, solar radiation and the minimum and maximum air temperatures were unavailable. Therefore we did not use weather generator for precipitation and we downloaded those unavailable climatic data from the Climate Forecast System Reanalysis (CFSR) during the time period 1979-2010 to calculate the statistical characteristics for weather generator. And we modeling the period 1991-2010 because the observed rainfall and flow data were available in that period. We think the last paragraph in section 2.2 has illustrated the usage of the observed rainfall data.

We suggest the following statement to illustrate the usage of the weather generator in the fourth paragraph in section 2.2:

"The SWAT model has developed a weather generator (WXGEN) to fill the missing climate data by the use of monthly statistics."

*Comment: How did you model land use management (.mgt) where did you obtain the data.*

Reply: We think that the land use management is not within the scope of this study. The land use management (.mgt) file contains input data for planting, harvest, irrigation applications, nutrient applications, pesticide applications, and tillage operations. We used the default setting for these operations in .mgt file.

*Comment: Please add table with data used in the model. For example refer to this manuscripts: Glavan, M., Ceglar, A. and Pintar, M., 2015. Assessing the impacts of climate change on water quantity and quality modelling in small Slovenian Mediterranean catchment - lesson for policy and decision makers. Hydrological Processes, 29(14): 3124-3144.*

Reply: Thanks for your good suggestion. The following table is added to the section 2.2-Model dataset:

| Data | Resolution | Source | Description |
|---|---|---|---|
| DEM | 90m×90m | http://srtm.csi.cgiar.org/ | Digital Elevation Model |
| Land use | 1km×1km | http://www.landcover.org/ | Land use classification |
| Soil | 30 arc-second | http://www.fao.org/soils-portal/soil-survey/soil-maps-and-databases/harmonized-world-soil-database-v12/en/ | Soil type classification and characterization of soil parameters |
| Global weather data | 30 stations | https://globalweather.tamu.edu/ | Relative humidity, wind speed, solar radiation and the minimum and maximum air temperatures |
| Observed rainfall | 138 gauges | Hydrologic Bureau of Huaihe River Commission | Daily data: 1991-2010; subdaily data: flood periods during 1991-2010 |
| Observed streamflow | 1 gauge | Hydrologic Bureau of Huaihe River Commission | Wangjiaba station, daily data for 1991-2010, sub-daily data for flood periods during 1991-2010 |

*Comment: 3.1 Development of: : : P5, L4: If you are following the method proposed by Jeong et al. (2010) please describe why and for what purpose was it made or used.*

Reply: Thank you for your valuable suggestion. We will explain specifically the model modification of Jeong et al. (2010) in the revised manuscript. I still need time to organize this part.

*Comment: 3.3 Model calibration Please introduce table with parameters used in calibration. Include also default value, range, final value. For example refer to this manuscripts: Glavan, M., Ceglar, A. and Pintar, M., 2015. Assessing the impacts of climate change on water quantity and quality modelling in*

*small Slovenian Mediterranean catchment - lesson for policy and decision makers. Hydrological Processes, 29(14): 3124-3144. This manuscript should also be part of introduction or discussion chapters as it clearly describes the process that need to be followed while using SWAT model.*

Reply: Thanks for your good suggestion and we would like to refer to the suggested manuscript. Since there are three calibrated parameter sets in this study (i.e., SWAT model, SWAT-EVENT model with basin level UH parameter, SWAT-EVENT model with sub-basin level UH parameter), we intend to add the calibrated values in the attachments. And the following table is added to section 3.3.1 to denote the model parameters:

| Parameters | Definition | lower bound | upper bound |
| --- | --- | --- | --- |
| ALPHA_BF | Baseflow alpha factor (days). | 0 | 1 |
| BIOMIX | Biological mixing efficiency. | 0 | 1 |
| BLAI | Maximum potential leaf area index. | 0 | 1 |
| CANMX | Maximum canopy storage (mm H2O). | 0 | 10 |
| CH_K(2) | Effective hydraulic conductivity in main channel alluvium (mm/hr). | 0 | 150 |
| CH_N | Manning's "n" value for the main channel. | 0 | 1 |
| CN2 | Initial SCS runoff curve number for moisture condition II. | -25 | 25 |
| EPCO | Plant uptake compensation factor. | 0 | 1 |
| ESCO | Soil evaporation compensation factor | 0 | 1 |
| GW_DELAY | Groundwater delay time (days). | -10 | 10 |
| GW_REVAP | Groundwater "revap" coefficient. | -0.036 | 0.036 |
| GWQMN | Threshold depth of water inthe shallow aquifer required for return flow to occur (mm H2O). | -1000 | 1000 |
| REVAPMN | Threshold depth of water in the shallow aquifer for "revap" or percolation to the deep aquifer to occur (mm H2O). | -100 | 100 |
| SMTMP | Snow melt base temperature (ºC). | 0 | 5 |
| SLOPE | Average slope | -25 | 25 |
| SLSUBBSN | Average slope length (m). | -25 | 25 |
| SMFMN | Melt factor for snow on December 21 (mm H2O/ºC-day). | 0 | 10 |
| SMFMX | Melt factor for snow on June 21 (mm H2O/ºC-day). | 0 | 10 |
| SMTMP | Snow melt base temperature (ºC). | -25 | 25 |
| SOL_ALB | Moist soil albedo. | -25 | 25 |
| SOL_AWC | Available water capacity of the soil layer (mm H2O/mm soil). | -25 | 25 |
| SOL_K | Saturated hydraulic conductivity (mm/hr). | -25 | 25 |
| SOL_Z | Depth from soil surface to bottom of layer (mm). | -25 | 25 |
| SURLAG | Surface runoff lag coefficient. | 0 | 10 |
| TIMP | Snow pack temperature lag factor. | 0 | 1 |
| TLAPS | Temperature lapse rate (ºC/km). | 0 | 50 |

*Comment: Please clearly describe what the scenarios were. I assume you had three scenarios as follows out from Table 3 where you presented for certain version (I assume SWAT-EVENT, please write*

*this in title of the table) three scenarios Daily simulation, Basin level UH parameter simulation and Sub-basin level UH parameter simulation. From Figure & I can see you had two scenarios Simulated daily discharge SWAT and simulated sub-daily discharge SWAT-EVENT. In methodologies clearly describe what is base scenario and to which scenario is it compared.*

Reply: Yes, three scenarios are:

(a) daily simulation with SWAT model;

(b) SWAT-EVENT model with basin level UH parameter ($t_{adj}$) for even-based simulation;

(c) SWAT-EVENT model with sub-basin level UH parameter ($t_{subadj}$) for event-based simulation.

We assume the referee is here referring Figure 6. This paper used a two-step comparison to prove that: (1) taking (a) as the base scenario and (b) as the compared scenario, temporal modification enabled the original SWAT model to simulate flood events and the improvements of the aggregated daily performances of the SWAT-EVENT model in Figure 6 were due to the higher temporal resolutions for input rainfall and the simulation time step; (2) taking (b) as the base scenario and (c) as the compared scenario, spatial modification improved the simulation accuracy for even-based floods (Table 3 and Figure 8).

The title of Table 3 is changed to "Performance evaluations for the daily with the SWAT model and sub-daily simulations for specific flood events with the SWAT-EVENT model"

We suggest to add a section in methodologies to describe the two-step comparison as follows:

"3.4 Improvement for even-based flood simulation

A two-step comparison was used to verify the improvement of the SWAT model for event-based flood simulation. Firstly, the aggregated daily results of the SWAT-EVENT model with default basin level UH parameter ($t_{adj}$) was compared to the original SWAT model to test the effectiveness of improvement at the temporal scale. Secondly, the SWAT-EVENT model with sub-basin level UH parameter ($t_{subadj}$) was compared to that with basin level UH parameter ($t_{adj}$) to assess the improvement effect at the spatial level."

*Comment: 5 Discussion P10, L28-30: Sentences from previous chapters are often repeated.*

Reply: We delete this part to avoid repeated.

*Comment: Conclusions P12, L16-30: All the text in the conclusions is just repeated from previous chapters. Delete existent text and please write down answers to this questions in conclusions: Why is this research unique? What are the shortcomings/uncertainties of this research? What did us and science*

***community learned from it? Future work?***

Reply: We will rework the whole conclusion section as follows:

"Flood forecasting is a synthetic system that integrates the data acquisition and processing, rainfall-runoff modeling and warning information release etc. Hydrological models are always the core part of the forecasting system. Model structures and model parameters are one of the most important issues for accurate flood forecasting (Noh et al., 2014). The original SWAT model was not competent to flood forecasting due to its initial design of long-term simulations with daily time-steps. This paper mainly focused on the modification of the structure of the original SWAT model to perform event-based simulation, which was applicable for the area without continuous long-term observations. The newly developed SWA-EVENT model was applied in the upper reaches of the Huaihe River. Model calibration and validation were made by the using of historical flood events, showing good simulation accuracy. To improve the spatial representation of the SWA-EVENT, the lumped UH parameters were then adjusted to the distributed ones. Calibration and validation results revealed the improvement of event-based simulation performances. This study expands the application of the original SWAT model in event-based flood simulation.

The determination of hydrological model parameters is an inevitable process before flood forecasting. Parameter estimations of distributed or semi-distributed hydrological models commonly depend on automated calibration procedure due to overparametrization. The optimal parameters of the SWAT-EVENT model were obtained by the automatic parameter calibration module that integrated SCE-UA algorithm in this study. However, serveral factors such as interactions among model parameters, complexities of spatio-temporal scales and statistical features of model residuals may lead to the parameter non-uniqueness, which is the source of the uncertainty in the estimated parameters. Uncertainty of model parameters will be finally passed to the model results, hence leading to certain risks in flood forecasting. In the future, emphasis will be placed on the quantification of the parameter uncertainty to provide better supports for flood operations.

Event-based runoff quantity and quality modeling has become a challenge task since the impact of hydrological extremes on the water quality is particularly important. The improvement of the SWAT model for event-based flood simulation will lay the foundation for dealing with the event-based water quality issues."

**References**

Jeong, J., Kannan, N., Arnold, J., Glick, R., Gosselink, L., and Srinivasan, R.: Development and Integration

of Sub-hourly RainfallRunoff Modeling Capability Within a Watershed Model, General Information, 24, 4505-4527, 2010.

Noh, S. J., Rakovec, O., Weerts, A. H., and Tachikawa, Y.: On noise specification in data assimilation schemes for improved flood forecasting using distributed hydrological models, J. Hydrol., 519, 2707-2721, 2014.

---

## Author Response (AR1)

**Response to Interactive comment Anonymous Referee #1**

We heartily appreciate the reviewer's assessment on this study and the valuable suggestions provided to improve this manuscript. We hereby provide our point by point responses how the comments by referee #1 will be addressed in the revised manuscript.

**1. Responses to major comments**

*Comment: The authors have to be more specific concerning the spatial resolution of the SWAT model. First, the SWAT model is a semi-distributed model consisting of subbasins and hydrological response units (HRU). Thus, please change from "distributed model" to "semi-distributed model".*

Reply: Thanks for your correction. The "distributed hydrological model" is changed to "semi-distributed hydrological model" in the manuscript.

*Comment: Since the manuscript is focused on spatial heterogeneity, a clear description of the three levels in the SWAT model, namely catchment, subbasin and HRU level, is required.*

Reply: We appreciate the comment. It is really necessary to state the spatial discretization of the SWAT model. The following statement is added to the third paragraph in section 1:

"To spatially characterize the inhomogeneity, the SWAT model delineates a catchment into a number of sub-basins, which were subsequently divided into Hydrologic Response Units (HRUs). In SWAT model, HURs are basic simulation units of the land phase of the hydrological cycle that controls the total yield of streamflow, sediment, pesticide and nutrient to the main channel in corresponding sub-basin. Afterwards the routing phase converges the land phase results to the watershed outlet through the channel network."

*Comment: Moreover, concerning the parameter heterogeneity and variability, it has to be clarified that some parameters (e.g. included in .bsn) are fixed for the whole catchment, others can be modified for each subbasins and a third group can be varied for each HRU. The authors should mention that the unit hydrograph is parameterized in the SWAT model version on the catchment level, which means that no spatial variation within a catchment is possible. In general, the SWAT model also allows a parameter variation on subbasin or HRU level. This has to be considered in the whole manuscript. In addition, the sentence in the abstract in L. 18 has to be more precise to avoid a misunderstanding.*

Reply: Thank you very much for your advice. We've realized that the parameter heterogeneity and variability are important issues in distributed or semi-distributed hydrological model application.

The following description is added to the fifth paragraph in section 1:

"Due to the spatial discretization in the SWAT model, the model parameters are categorized into three levels: (1) basin level parameters are fixed for the whole catchment; (2) sub-basin level parameters are varied with sub-basins; (3) HRU level parameters are distributed in different HRUs."

The sentence "In addition, the SWAT model provides a uniform parameter set with which to adjust the shape of the UH in each sub-basin." in the fifth paragraph in section 1 is rephrased as follows:

"By default, the UH related parameters in the SWAT model are on the basin level, which indicates that no spatial variation within a catchment is possible when adjusting the shape of the UH in each sub-basin.".

The sentence in the abstract in L. 18 is changed as follows:

"SWAT uses a basin level parameter that is fixed for the whole catchment to parameterize the Unit Hydrograph (UH), thereby ignoring the spatial heterogeneity among the sub-basins when adjusting the shape of the UHs."

*Comment: The SWAT model version 2005 is a very old one. SWAT2009 and SWAT2012 are available since several years. Thus, please justify the use of SWAT2005. This is in particular relevant since the SWAT model was continuously improved and bugs were removed. Thus, the newer versions are certainly better. Please give a statement on this.*

*The justification for the use of SWAT2005 (P.3, L. 13-15) cannot be accepted. Certainly it is possible to use SWAT-CUP for calibration, but it is certainly possible to use different calibration method for all SWAT version. There are lots of study using a different calibration approach for the SWAT model. Moreover a link between the use of SWAT2005 and the selected study catchment is not clear. Thus, please remove this part and provide a better explanation why SWAT2005 was selected instead of SWAT2009 or SWAT2012.*

Reply: We propose to add the following for justification at the last paragraph in section 1:

"SWAT is an open-source code model, which makes it possible to produce such a modification. The SWAT2005 version has an existing auto-calibration module and such integrated design of model simulation and auto-calibration is easily manageable and modified since there is no need to couple external optimization algorithms."

*Comment: It is not acceptable to use only four sub-basins for 30630 km² catchment and claim at the same time limitations in spatial heterogeneity. According to my experience at least more than 100*

*subbasins can be expected for this catchment size. In particular, since the manuscript is focused on spatial heterogeneity, it is surprising that the subbasin number is very low.*

Reply: We agree with the reviewer for this comment. Sub-basin is assumed homogeneous with parameters at the sub-basin level. Since this paper discussed the spatial differences in model parameter, we are going to redefine the sub-basins and do all the simulations again. In the revised manuscript, we add the following statement to describe the watershed delineation in the second paragraph in section 2.2:

"Since there is no specific instruction to subdivide the catchment, the threshold sub-basin size was decided by the model developer, depending on the computational time and the size of the catchment (Romanowicz et al., 2005). Consequently, the study catchment was divided into 21 sub-basins according to the given threshold of 844.64 km$^2$, as shown in Fig. 1."

*Comment: Moreover, the SWAT model only provides spatially located outputs for each subbasin. In contrast, the authors stated that there are 138 gauges available. Thus, why do you not define a separate subbasin for each gauge or at least for the majority of the gauges? This would be even more a good approach to consider spatial heterogeneity.*

Reply: Since we consider redefinition of the sub-basins, we will use the Taisen Polygon Method to calculate area rainfall in each sub-basin to consider the spatial heterogeneity of rainfall gages.

*Comment: The evaluation of the model results with Nash-Sutcliffe efficiency coefficient (ENS) and coefficient of determination (R2) is redundant. Both indices are mathematically closely related. R2 did not provide any additional knowledge about process or parameter behaviour. Even though that I am aware that there are still publications using ENS and R2, it is not anymore state-of-the-art. At the same time, the use of three performance criteria is recommended. Thus, please select in addition to ENS and PBIAS, a contrasting performance criteria which provides additional information and replace R2.*

Reply: We appreciate the comment. We would like to use the ratio of the root mean square error to the standard deviation of measured data ($R_{SR}$) instead of $R^2$. The $R_{SR}$ index standardizes the root mean square error using the observations standard deviation, varying from 0 to a positive value. The optimal value of $R_{SR}$ is 0, which indicates the perfect model simulation."

*Comment: I do not think that your approach really shows an example for flood forecasting. It is a sub-daily model studies, but I do not see that there is a forecasting. The model is calibrated and validated.*

*Could you please provide more information on how your approach is related to flood forecasting? Or say that this approach might be also beneficial for flood forecasting, but this was not considered here or will be part of future projects?*

Reply: Thanks for your comment. We want to make some explanations here.

The hydrological model itself is not an example for flood forecasting, but a core part of the flood forecasting system. The quality of the hydrological model (in terms of its structure and parameter estimates) is one of the important factors for accurate flood forecasting (Noh et al., 2014). Thus this paper mainly focused on the modification of the structure of the original SWAT model to verify its suitability for flood simulation in study area, which is tightly related to the flood forecasting.

Meantime, model parameter estimation is an inevitable issue accompanied by the application of the hydrological model. Typically, calibration is performed by using multiple historical flood events data. Subsequently, the model validation consists in running the model under another group of historical flood events, with the input of parameter values thus being estimated in the calibration phase. This kind of calibration and validation is the common solution in many flood forecasting practices (Hapuarachchi and Wang, 2008). Thus the modified SWAT model was calibrated with 16 flood events from 1991 to 2004 and validated with 8 flood events from 2005 to 2010 in this paper.

In addition, Berthet et al. (2009) declared that the major drawback of the continuous simulation lies in its data requirements: long continuous precipitation time series up to the day of interest are difficult to obtain in an operational forecasting perspective. Yao et al. (2014) claimed that long continuous daily simulations are implemented to compute the soil moisture states that are used as antecedent conditions for the flood events in the operational use. Therefore the Figure 3 in this paper showed the operation at two time scales (i.e., continuous daily simulation and event-base sub-daily simulation).

In summary, this study addressed the model-based problems that related to flood forecasting. However we think it is still necessary to further clarify the relationship between the flood forecasting and our approaches.

The second paragraph in section 1: "A large number of distributed or semi-distributed hydrological models have been applied in flood forecasting (BEVEN and KIRKBY, 1979;Singh, 1997;Xiong and Guo, 2004;Mendes and Maia, 2017;Hapuarachchi et al., 2011)." replaces the "The Soil and Water Assessment Tool (SWAT) model (Arnold et al., 1998;Srinivasan et al., 1998;Arnold and Fohrer, 2005) is the most widely

used of the prevailing distributed models." to emphasize the model-based approaches of flood forecasting.

We propose a paragraph explaining the approaches that have addressed the model structures and parameters issues in flood forecasting in the conclusions section:

"Flood forecasting is a synthetic system that integrates the data acquisition and processing, rainfall-runoff modeling and warning information release etc. Hydrological models are always the core part of the forecasting system. Model structures and model parameters are one of the most important issues for accurate flood forecasting (Noh et al., 2014). The original SWAT model is not competent to flood forecasting due to its initial design of long-term simulations with daily time-steps. This paper mainly focused on the modification of the structure of the original SWAT model to perform event-based simulation, which was applicable for the area without continuous long-term observations. The newly developed SWA-EVENT model was applied in the upper reaches of the Huaihe River. Model calibration and validation were made by the using of historical flood events, showing good simulation accuracy. To improve the spatial representation of the SWA-EVENT, the lumped UH parameters were then adjusted to the distributed ones. Calibration and validation results revealed the improvement of event-based simulation performances. This study expands the application of the original SWAT model in event-based flood simulation."

*Comment: I have one general major comment: The authors suggest to improve the SWAT model in two ways. At first, at the spatial level, the model parameter t_adj is moved from the catchment level to the subbasin level to allow an individual parameterizsation for each subbasin and thus to consider spatial heterogeneity. Secondly, at the temporal scale, a sub-daily modeling is suggested to improve the representation of flood peaks. Both aspects, spatial and temporal improvements, are not clearly enough separated. It would interesting to know why aspect improves the model more and in which part of the hydrograph. The study would benefit from a four-step comparison instead of a two-step comparison. To be more precise: I recommend to add two cases: (1) Sub-daily calculation with t_adj at catchment level (without t_subadj) and the opposite case (2) daily calculation with t_adj at subbasin level (with t_subadj).*

Reply: Thanks for your comment. We assume the referee is here referring the four-step comparison:

daily calculation with catchment level parameter ($t_{adj}$);

daily calculation with sub-basin level parameter ($t_{subadj}$);

sub-daily calculation with catchment level parameter ($t_{adj}$);

sub-daily calculation with sub-basin level parameter ($t_{subadj}$).

If so, we want to respectfully clarify some confusions. In SWAT model, the Unit Hydrograph (UH) method is only used for sub-daily simulation, rather than a daily simulation. The main reason of this situation is that the flood hydrograph resulting from a known storm often vary significantly within sub-daily time-scales, while the daily calculation may exceed the time of concentration for most of the sub-basins. Even for large sub-basins with a time of concentration greater than 1 day, SWAT has incorporated a lag equation to store a portion of the surface runoff release to the main channel. Thus we think there are no such cases: daily calculation with basin level parameter ($t_{adj}$) and daily calculation with sub-basin level parameter ($t_{subadj}$).

The other argument is that modification at the temporal level was the first step, and modification at the spatial level was the second step.

At the temporal level, there are two drawbacks in the application of the original SWAT model for event-based flood simulation: (1) algorithms with daily time step for some hydrological processes are implicitly assumed (2) the continuous long-term simulation loop of its initial design. This paper referenced the methodologies in a previous study (Jeong et al., 2010) to solve the aforementioned problem (1). Then this paper broke down the continuous cycle of the model structure to solve the problem (2). With this, the SWAT-EVENT model was developed to simulate the event-based floods. At the spatial level, the UH parameter was modified from basin level to sub-basin level to represent the spatial heterogeneity of studied catchment, expecting a more reasonable UH characterization in the SWAT-EVENT model.

This paper used a two-step analysis to prove the improvement of the SWAT model for event-based flood simulation. Firstly, as noted in section 4.2, the newly developed SWAT-EVENT model simulated with the refined sub-basin level UH parameter ($t_{subadj}$). Nash-Sutcliffe efficiency coefficient ($E_{NS}$), relative peak discharge error ($E_{RP}$), relative peak time error ($E_{RPT}$) and relative runoff volume error ($E_{RR}$) were used to evaluate the applicability of the SWAT-EVENT model for event-based flood simulation. In addition, SWAT-EVENT model results were also compared to the daily simulation with the SWAT model to verify its superiority of the simulation in flood seasons. Secondly, as noted in section 4.3, in order to analyze the influences of UH parameters on the SWAT-EVENT model performances, the lumped parameter ($t_{adj}$) was then calibrated while the other parameters remained unchanged exactly as the distributed case was calibrated. At this stage, the SWAT-EVENT model was simulated with different UH methods while keeping other modeling conditions consistent, the changes in simulation results would be directly attributable to the UH

parameters.

*Comment: The model modification of Jeong et al. (2010) to simulate on sub-daily resolution needs to be explained and not only mentioned (P.5, L.4). This is a core point of the manuscript. The readers need to understand this modification without reading the paper of Jeong et al. (2010).*

Reply: Thank you for your valuable suggestion. Model modification of Jeong et al. (2010) is explained in the revised manuscript in the second paragraph in section 3.1.

*Comment: You have mentioned that the SWAT is in its default version not adapted to sub-daily flood peak simulations. Keeping in mind that a large number of models is available: Why do you have selected the SWAT model and not a model which is focused on the hydrograph simulation. The major points of the SWAT model such as nutrient simulation, detailed land managements operation etc. are not relevant for your study.*

Reply: The perceptive comment shows the reviewer's knowledge in the field, and we have to admit that the SWAT model is not the first choice of flood simulation because there are so many other models have good applicability in this field. Here we respectfully argue that this study still has certain scientific significance. Though the highlights of the model of the SWAT model are the predictions of the impact of land management practices on water, sediment and agricultural chemical yields, runoff simulation is always a fundamental. Moreover, we have noticed that the study on the event-based water quality assessment has been a hot topic (He et al., 2010;Nguyen and Meon, 2013;He et al., 2011). Therefore the improvement of the SWAT model for event-based flood simulation will lay the lay the foundation for event-based water quality modeling. To emphasize our points, the following statement is added to the conclusion section:

"Event-based runoff quantity and quality modeling has become a challenge task since the impact of hydrological extremes on the water quality is particularly important. The improvement of the SWAT model for event-based flood simulation will lay the foundation for dealing with the event-based water quality issues."

*Comment: The model results from SWAT2005 are used as input for SWAT-EVENT to simulation the flood peak. Is the model output of SWAT-EVENT then transfered back to SWAT2005? This point might be relevant since the first two flood events occured with a time lag of 19 days (P.9, L. 11). Thus, I expect a difference in the model states at the end of the first flood between SWAT and SWAT-EVENT. In this context, I like to mention that SWAT-EVENT does not impact the amount of water available in the system,*

*but the water redistribution.*

Reply: The answer to the question "Is the model output of SWAT-EVENT then transfered back to SWAT2005" is no. In fact, the daily SWAT model and the sub-daily SWAT-EVENT model were executed independently. According to Figure 3, the continuous daily SWAT model ran first to calculate the antecedent conditions for each flood events. After this, the SWAT-EVENT model began to run. Such continuous soil moisture modeling using the daily data series to estimate sub-daily initial conditions would be a traditional solution for the derivation of the antecedent moisture conditions, as suggested by Nalbantis (1995).

*Comment: P.5, L.16-29: This part needs to be reformulated to present the idea in a better way. In the current version, it is difficult to understand.*

Reply: We agree that better reformulation about the modification of the SWAT model should be presented. This part has been removed. Reorganization of this part is as follow:

"However, the event-based modeling requires a separate method to derive the antecedent conditions of model states. The combination of daily continuous modeling and sub-daily event-based modeling was used in this study (Fig. 3). A continuous daily rainfall sequence was imported into the original SWAT model to independently perform long-term daily simulations. In the SWAT model, there are another two subroutines "varinit" and "rchinit" initializing the daily simulation variables for the land phase of the hydrologic cycle and the channel routing, respectively. In the SWAT-EVENT model, condition judgments were added into those two initialization subroutines. That is, when the simulation process is at the beginning of a given flood event, antecedent soil moisture and antecedent reach storage are set equal to the respective values extracted from the long-term daily simulations of the original SWAT model; otherwise, they should be updated by the SWAT-EVENT model simulation states of the previous day."

*Comment: P. 7, L. 1-2: This is a major point of the manuscript and has to be emphasised. A new model parameter is introduced at the subbasin level to include spatial heterogeneity. This is really important that it becomes clear.*

Reply: Thank you for the suggestion. We think it is really necessary to add more descriptions for this major point. The last paragraph in section 3.2 is changed to:

"According to Eq. (2), the time base of the UH ($t_b$) is determined by both concentration time for the sub-basin ($t_c$) and shape adjustment factor ($t_{adj}$) concurrently. As seen in Fig. 1 and Table 2, there are obvious spatial differences of the geographical attributes among sub-basins. For instance, the values of

sub-basin area vary from 2.94 km$^2$ to 4795.46 km$^2$ with the average value of 1437.12 km$^2$, and the mean slopes in source sub-basins (e.g. sub 1, sub 16, sub 19, sub 20 and sub 21) are much steeper than those in downstream sub-basins (e.g. sub 7, sub 8 and sub 11). As a result, the sub-basin concentration time $t_c$ synthesizes all those geographical attributes and it can fully present the spatial differences among sub-basins according to Eq. (5) and (6). However, the parameter $t_{adj}$ in Eq. (2) is a basin level parameter possessing a lumped value for all sub-basins, meaning that the spatial heterogeneity of $t_b$ may be homogenized due to the constraints between sub-basins. Generally, the time base of triangular UH ($t_b$) should be reduced to produce increased peak flow for steep and small sub-basins, or increased to produce decreased peak flow for flat and large sub-basins. Thus, the shape adjustment parameter $t_{adj}$ was modified from the basin level to the sub-basin level, and renamed $t_{subadj}$ which allowed the UHs to be adjusted independently by distributed values."

*Comment: P. 11, L. 18-19: This statement is not right. The SWAT model is not limited in representing low flows. It is more that there is a trade-off between high and low flow simulations at the same time. At it is true that it is difficult to represent high and low flows in a very good quality with the same model run. This is by the way an often occurring problem in hydrological modelling. The major point here is that the selection of the performance measures is at the same time a decision on the study focus. By selecting the NashSutcliffe Efficiency high flows are more weighted than low flows. Thus, it would not be a surprise if the high flows are well represented while low flows perform poorly. This results could be different if using ENSlog or a different performance measure related to low flows. Please improve this statement.*

Reply: Thank you very much for pointing out my mistake. The following statement is going to correct the mistake:

"On the one hand, the SWAT model was calibrated using the sum of squares of the residuals as the objective function, which was more sensitive to high flows than low flows. Thus the calibration results ensured the simulation accuracy at the expense of the low flow performances"

*Comment: The aspect of flood forecasting is strongly emphasised in the conclusion. I still do not see that the strength of the manuscript is related to flood forecasting. Please rework the conclusion accordingly.*

Reply: We will rework the whole conclusion section as follows:

"Flood forecasting is a synthetic system that integrates the data acquisition and processing, rainfall-runoff modeling and warning information release etc. Hydrological models are always the core part of the forecasting system. Model structures and model parameters are one of the most important issues for accurate flood forecasting (Noh et al., 2014). The original SWAT model was not competent to flood forecasting due to its initial design of long-term simulations with daily time-steps. This paper mainly focused on the modification of the structure of the original SWAT model to perform event-based simulation, which was applicable for the area without continuous long-term observations. The newly developed SWA-EVENT model was applied in the upper reaches of the Huaihe River. Model calibration and validation were made by the using of historical flood events, showing good simulation accuracy. To improve the spatial representation of the SWA-EVENT, the lumped UH parameters were then adjusted to the distributed ones. Calibration and validation results revealed the improvement of event-based simulation performances. This study expands the application of the original SWAT model in event-based flood simulation.

The determination of hydrological model parameters is an inevitable process before flood forecasting. Parameter estimations of distributed or semi-distributed hydrological models commonly depend on automated calibration procedure due to overparametrization. The optimal parameters of the SWAT-EVENT model were obtained by the automatic parameter calibration module that integrated SCE-UA algorithm in this study. However, serveral factors such as interactions among model parameters, complexities of spatio-temporal scales and statistical features of model residuals may lead to the parameter non-uniqueness, which is the source of the uncertainty in the estimated parameters. Uncertainty of model parameters will be finally passed to the model results, hence leading to certain risks in flood forecasting. In the future, emphasis will be placed on the quantification of the parameter uncertainty to provide better supports for flood operations.

Event-based runoff quantity and quality modeling has become a challenge task since the impact of hydrological extremes on the water quality is particularly important. The improvement of the SWAT model for event-based flood simulation will lay the foundation for dealing with the event-based water quality issues."

**2. Responses to specific comments**

*Comment: P.3, L.3: How do you "relate hydrologic response to specific catchment characteristics"?*

*By parameter settings?*

Reply: The dimensionless UH used in the SWAT model is just one form of the Synthetic Unit Hydrograph (SUH), which can be used to the ungauged catchments. The SUH was derived from catchment characteristics rather than rainfall-runoff data (Bhunya, 2011). According to Equation (3), the UH was defined based on the hydrologic property of the catchment.

*Comment: P.4, L. 18: The weather generator is only used in the case of missing climate data. Please improve this statement.*

Reply: We suggest the following statement to be a replacement:

"The SWAT model has developed a weather generator (WXGEN) to fill the missing climate data by the use of monthly statistics."

*Comment: P. 5, L. 21: It was not mentioned before that the SWAT model includes HRUs. Please improve the description of the SWAT model.*

Reply: We suggest the following description to be added to the third paragraph in section 1:

"To spatially characterize the inhomogeneity, the SWAT model delineates a catchment into a number of sub-basins, which were subsequently divided into Hydrologic Response Units (HRUs). In SWAT model, HURs are basic simulation units of the land phase of the hydrological cycle that controls the total yield of streamflow, sediment, pesticide and nutrient to the main channel in corresponding sub-basin. Afterwards the routing phase converges the land phase results to the watershed outlet through the channel network."

*Comment: P. 7, L. 9: Please denote the 26 parameters, maybe in the attachments.*

Reply: Thanks for the good suggestion. We denote the model parameters in Appendix A.

*Comment: P. 8, L.13: Please explain the three indices, at best with equations.*

Reply: We explain the three indices with equations (7), (8) and (9).

Reply: Thanks for the comments. We want to make some explanations here. Weather generator is only used in the case of missing climate data. In this study only the observed rainfall data were available while the other climatic data such as relative humidity, wind speed, solar radiation and the minimum and maximum air temperatures were unavailable. Therefore we did not use weather generator for precipitation and we downloaded those unavailable climatic data from the Climate Forecast System Reanalysis (CFSR) during the time period 1979-2010 to calculate the statistical characteristics for weather generator. And we modeling the period 1991-2010 because the observed rainfall and flow data were available in that period. We think the last paragraph in section 2.2 has illustrated the usage of the observed rainfall data.

We suggest the following statement to illustrate the usage of the weather generator in the fourth paragraph in section 2.2:

"The SWAT model has developed a weather generator (WXGEN) to fill the missing climate data by the

use of monthly statistics."

*Comment: How did you model land use management (.mgt) where did you obtain the data.*

Reply: We think that the land use management is not within the scope of this study. The land use management (.mgt) file contains input data for planting, harvest, irrigation applications, nutrient applications, pesticide applications, and tillage operations. We used the default setting for these operations in .mgt file.

*Comment: Please add table with data used in the model. For example refer to this manuscripts: Glavan, M., Ceglar, A. and Pintar, M., 2015. Assessing the impacts of climate change on water quantity and quality modelling in small Slovenian Mediterranean catchment - lesson for policy and decision makers. Hydrological Processes, 29(14): 3124-3144.*

Reply: Thanks for your good suggestion. The following table is added to the section 2.2-Model dataset:

| Data | Resolution | Source | Description |
|---|---|---|---|
| DEM | 90m×90m | http://srtm.csi.cgiar.org/ | Digital Elevation Model |
| Land use | 1km×1km | http://www.landcover.org/ | Land use classification |
| Soil | 30 arc-second | http://www.fao.org/soils-portal/soil-survey/soil-maps-and-databases/harmonized-world-soil-database-v12/en/ | Soil type classification and characterization of soil parameters |
| Global weather data | 30 stations | https://globalweather.tamu.edu/ | Relative humidity, wind speed, solar radiation and the minimum and maximum air temperatures |
| Observed rainfall | 138 gauges | Hydrologic Bureau of Huaihe River Commission | Daily data: 1991-2010; subdaily data: flood periods during 1991-2010 |
| Observed streamflow | 1 gauge | Hydrologic Bureau of Huaihe River Commission | Wangjiaba station, daily data for 1991-2010, sub-daily data for flood periods during 1991-2010 |

*Comment: 3.1 Development of: : : P5, L4: If you are following the method proposed by Jeong et al. (2010) please describe why and for what purpose was it made or used.*

Reply: Thank you for your valuable suggestion. Model modification of Jeong et al. (2010) is explained in the revised manuscript in the second paragraph in section 3.1.

*Comment: 3.3 Model calibration Please introduce table with parameters used in calibration. Include also default value, range, final value. For example refer to this manuscripts: Glavan, M., Ceglar, A. and Pintar, M., 2015. Assessing the impacts of climate change on water quantity and quality modelling in small Slovenian Mediterranean catchment - lesson for policy and decision makers. Hydrological Processes, 29(14): 3124-3144. This manuscript should also be part of introduction or discussion chapters as it clearly describes the process that need to be followed while using SWAT model.*

Reply: Thanks for your good suggestion. We denote the model parameters in Appendix A.

*Comment: Please clearly describe what the scenarios were. I assume you had three scenarios as follows out from Table 3 where you presented for certain version (I assume SWAT-EVENT, please write this in title of the table) three scenarios Daily simulation, Basin level UH parameter simulation and Sub-basin level UH parameter simulation. From Figure & I can see you had two scenarios Simulated daily discharge SWAT and simulated sub-daily discharge SWAT-EVENT. In methodologies clearly describe what is base scenario and to which scenario is it compared.*

Reply: Yes, three scenarios are:

(a) daily simulation with SWAT model;

(b) SWAT-EVENT model with sub-basin level UH parameter (tsubadj) for even-based simulation;

(c) SWAT-EVENT model with basin level UH parameter (tadj) for event-based simulation while keeping the other model parameters in line with the scenario (b).

We assume the referee is here referring Figure 6. This paper used a two-step analysis to prove the improvement of the SWAT model for event-based flood simulation. Firstly, as noted in section 4.2, the newly developed SWAT-EVENT model simulated with the refined sub-basin level UH parameter ($t_{\text{subadj}}$). Nash-Sutcliffe efficiency coefficient ($E_{\text{NS}}$), relative peak discharge error ($E_{\text{RP}}$), relative peak time error ($E_{\text{RPT}}$) and relative runoff volume error ($E_{\text{RR}}$) were used to evaluate the applicability of the SWAT-EVENT model for event-based flood simulation. In addition, SWAT-EVENT model results were also compared to the daily simulation with the SWAT model to verify its superiority of the simulation in flood seasons. Secondly, as noted in section 4.3, in order to analyze the influences of UH parameters on the SWAT-EVENT model performances, the lumped parameter ($t_{\text{adj}}$) was then calibrated while the other parameters remained unchanged exactly as the distributed case was calibrated. At this stage, the SWAT-EVENT model was simulated with different UH methods while keeping other modeling conditions consistent, the changes in simulation results would be directly attributable to the UH parameters.

*Comment: 5 Discussion P10, L28-30: Sentences from previous chapters are often repeated.*

Reply: We delete this part to avoid repeated.

*Comment: Conclusions P12, L16-30: All the text in the conclusions is just repeated from previous chapters. Delete existent text and please write down answers to this questions in conclusions: Why is this research unique? What are the shortcomings/uncertainties of this research? What did us and science*

*community learned from it? Future work?*

Reply: We will rework the whole conclusion section as follows:

[revised manuscript text omitted]

批注 [L37]: Since the catchment was re-delineated, this table was changed.

| Sub-basin No. | Drainage area (km$^2$) | Mean elevation (m) | Mean slope (°) | Mean slope length (m) | Longest tributary length (km) | Average slope of the tributary (m m$^{-1}$) |
|---|---|---|---|---|---|---|
| 1 | 1997.74 | 83 | 7.49 | 60.96 | 140.06 | 0.0010 |
| 2 | 262.15 | 62 | 1.05 | 121.91 | 49.46 | 0.0001 |
| 3 | 1032.38 | 60 | 1.41 | 121.91 | 130.46 | 0.0010 |
| 4 | 2515.71 | 161 | 4.58 | 91.44 | 175.31 | 0.0040 |
| 5 | 1712.57 | 42 | 1.20 | 121.91 | 121.25 | 0.0010 |
| 6 | 3852.86 | 57 | 2.71 | 91.44 | 295.11 | 0.0010 |
| 7 | 4.26 | 30 | 1.32 | 121.91 | 4.13 | 0.0010 |
| 8 | 722.28 | 32 | 0.93 | 121.91 | 81.10 | 0.0001 |
| 9 | 2.94 | 32 | 2.26 | 91.44 | 4.92 | 0.0020 |
| 10 | 927.36 | 49 | 0.95 | 121.91 | 101.10 | 0.0010 |
| 11 | 450.41 | 31 | 1.12 | 121.91 | 73.08 | 0.0001 |
| 12 | 31.34 | 35 | 1.59 | 121.91 | 16.31 | 0.0010 |
| 13 | 477.56 | 47 | 0.88 | 121.91 | 48.86 | 0.0001 |
| 14 | 295.68 | 49 | 1.13 | 121.91 | 42.90 | 0.0010 |
| 15 | 886.69 | 54 | 1.10 | 121.91 | 104.65 | 0.0010 |
| 16 | 4795.46 | 96 | 7.28 | 60.96 | 209.67 | 0.0020 |
| 17 | 999.62 | 57 | 3.68 | 91.44 | 95.88 | 0.0040 |
| 18 | 2216.48 | 50 | 4.43 | 91.44 | 141.88 | 0.0030 |
| 19 | 2029.25 | 148 | 13.17 | 24.38 | 170.84 | 0.0040 |
| 20 | 2399.24 | 74 | 8.42 | 60.96 | 160.71 | 0.0060 |
| 21 | 2567.61 | 100 | 8.80 | 60.96 | 120.53 | 0.0060 |

**Table 3 SWAT model performance statistics for the calibration and validation periods.**

|  | $E_{NS}$ | $R_{SR}$ | $P_{BIAS}$ (%) |
|---|---|---|---|
| Calibration | 0.80 | 0.45 | -14.32 |
| Validation | 0.83 | 0.42 | -18.29 |

批注 [L38]:

**Table 4 Performance evaluations for the daily simulation with the SWAT model for specific flood events, and the SWAT-EVENT model performances with sub-basin level UH parameters and basin level UH parameter.**

[revised manuscript text omitted]

---

## Author Response (AR2)

**Response to Interactive comment Anonymous Referee #1**

We heartily appreciate the reviewer's assessment on this study and the valuable suggestions provided to improve this manuscript. We have studied comments carefully and have made correction which we hope meet with approval. We hereby provide our point by point responses how the comments by referee #1 will be addressed in the revised manuscript.

**1. Responses to major comments**

*Comment: In this study, it is shown how to improve the SWAT model for flood peaks. This is a justification for a study. However, I still do not see a strong link to flood forecasting in this article. You may argue that reasonable flood peak simulations are required for flood forecasting (a point that can be raised e.g. in the discussion), but to use flood forecasting in the title, I expect a higher contribution to this topic.*

Reply: Thanks for the comments. We have realized that the significance of flood forecasting is to provide early warnings of the forthcoming floods, thus, real-time flood modeling and prediction should be emphasized. However this article was about the hydrological simulation with the historical data. Strictly speaking, it was an application of the flood simulation rather than the flood forecasting. For this concern, the article title was changed from "Improvement of the SWAT model for event-based flood forecasting on a sub-daily time scale" to "Improvement of the SWAT model for event-based flood simulation on a sub-daily time scale".

*Comment: I am still not convinced that spatial heterogeneity is adequately considered. The high number of rainfall gauges might allow a higher number of subbasins. I see that the number of subbasins increased from 4 to 21 compared to the first version. However, I think that some statements of the spatial heterogeneity within the subbasins would be helpful to justify the still relatively low number of subbasins. This is in particular relevant since the SWAT model is based on the HRU concept (spatially not located hydrotops) and that a focus of the article is the calculation of the time of concentration. A higher spatially explicit discretization would certainly be helpful in calculating the time of concentration realistically.*

Reply: Thanks for the valuable suggestion. As reviewer suggested that the high number of rainfall gauges might allow a higher number of subbasins, the catchment was further divided into 136 subbasins.

*Comment: I am not satisfied with the parameter selection and the method presented for sensitivity*

*analysis and model calibration (see detailed comments below).*

Reply: Parameter selection, sensitivity analysis and model calibration presentations were reworked according to corresponding comments.

*Comment: I have one comment to the calculation of objective functions for SWAT-EVENT and the comparison with the classical SWAT model. To calibrate the SWAT, the entire time series was used. For SWAT-EVENT, only events were used. It is then not very surprising that an optimization of a smaller time period leads to better results within this time period. A fair comparison would be to calibrate the SWAT model also only for the event period. Maybe I understand something wrong, but currently it seems to be that the better result in SWAT-EVENT is (at least partly) related to the selection of a smaller time period. In all cases, the study would benefit from showing the difference between SWAT and SWAT-EVENT by using the same event time period.*

Reply: It is really true as reviewer suggested that same simulation period should be selected for both model calibrations. Therefore, daily simulation with the SWAT model only for event periods was added. Then a fair comparison was obtained by comparing the SWAT daily simulation results to the aggregated SWAT-EVENT simulation results.

**2. Responses to comments**

*Comment: P. 1, L.15-16: I agree that SWAT is a good water quality model, but needs time for several simulations due to its complexity. Thus, there is an explanation missing why you have selected the SWAT model for flood studies. A simpler but also semi-distributed model which is only focused on hydrology might be more useful.*

Reply: Thanks for the comment. It is true that hydrology modeling, rather than water quality modeling, is always the major concern for flood studies. Thus, the advantage of the SWAT model in water quality modeling was not that significant in such application. However, in the revised manuscript, we emphasized the SWAT model's good applicability in dealing with management practices and changed scenarios occurring in the watershed. This is of great importance to the pre-assessment of flood control measures. Considering the reviewer's suggestion, the following explanation of the SWAT model used for this article was added in P1, L15-17:

A model that is capable of predicting the hydrological responses in watershed with management practices during flood periods would be a crucial tool for pre-assessment of flood reduction measures. The Soil and Water Assessment Tool (SWAT) is a semi-distributed hydrological model that is well capable of runoff and water quality modeling under changed scenarios.

For further explanation the usage of the SWAT model, the following contents were also added in P2, L29-P3, L6:

Among many distributed (semi-distributed) models, the one that is capable of predicting the hydrological responses in watersheds with management practices would provide scientific reference for preventing flood and mitigating its adverse effects.

Soil and Water Assessment Tool (SWAT) model (Kiniry et al., 2005) is a typical semi-distributed hydrological model that delineates a catchment into a number of sub-basins, which were subsequently divided into Hydrologic Response Units (HRU) representing the unique combination of land cover, soil type, and slope class within a sub-basin. SWAT model integrated well with Geographic Information System (GIS), having great potential in dealing with spatial flood control measures. In addition, the SWAT model is widely applied for runoff and water quality modeling under changed scenarios (Glavan et al., 2015;Yu et al., 2018;Qiu et al., 2017;Baker and Miller, 2013;Yan et al., 2013).

*Comment: P.2., L.9: For flood forecasting purpose, it is required to simulate on time and not only the past. At least in the future perspective, on-time simulation is needed. Please refer to this point (e.g. in the discussion).*

Reply: We have already changed the topic from flood forecasting to flood simulation. So the original part relating to the flood forecasting description would be replaced by the following contents in P2, L9-14:

Floods have caused enormous losses to economy, society and ecological environment around the world (Doocy et al., 2013;Werritty et al., 2007;Guan et al., 2015). China is a flood-prone country, which suffers from severe flooding almost every year (Zhang et al., 2002). In this situation, protection against flooding has always been the government's primary task that brooks no delay. A series of structural and non-structural flood mitigation measures have been conducted to control and manage the floods (Guo et al., 2018). However, accurate flood simulations would be particularly important for such design- or management-related issues.

*Comment: P.2., L.18: Here, a justification for the model selection is missing. There are a huge amount of spatial-distributed model. At least a few words on that are required.*

Reply: Thanks for your valuable advice. We explained several spatial-distributed hydrological models, such as the TOPMODEL, VIC, HBV and HEC-HMS in P2, L24-29 as follows:

Beven et al. (1984) firstly tested the applicability of the TOPMODEL in flood simulation for three U.K. catchments and suggested that the model could be a useful approach for ungauged catchments. Variable Infiltration Capacity (VIC) model is also playing an increasing role in flood simulation (Wu et al., 2014;Yigzaw and Hossain, 2012). The applications of the HBV model for flood simulation could be found in many studies (Haggstrom et al., 1990;Grillakis et al., 2010;Kobold and Brilly, 2006). HEC-HMS model was able to provide reasonable flood simulation results in the San Antonio River Basin (Ramly and Tahir, 2016). Among many distributed (semi-distributed) models, the one that is capable of predicting the hydrological responses in watersheds with management practices would provide scientific reference for preventing flood and mitigating its adverse effects.

*Comment: P.2., L. 18: Please cite the SWAT model here with the original reference.*

Reply: Thanks for the reminding. The original reference of the SWAT model was added in P3, L1.

*Comment: P.2, L.26: What is meant here with „by research involving model parameters"? Please be precise. Model parameters themselves are not a research topic. Parameter sensitivity, optimization, screening etc would be a topic.*

Reply: Thanks for the correction. This part has been deleted. To state the applications of the SWAT model in many studies, the following content was added in P3, L5-6:

In addition, the SWAT model is widely applied for runoff and water quality modeling under changed scenarios (Glavan et al., 2015;Yu et al., 2018;Qiu et al., 2017;Baker and Miller, 2013;Yan et al., 2013)

*Comment: P.3., L. 18: How is flood forecasting included?*

Reply: More precisely, the "flood forecasting" was changed to "flood simulation".

*Comment: P.3, L.20: All SWAT model versions include an auto-calibration routine. This is not specific for SWAT2005. I can understand that you prefer to use SWAT2005 because you have reasons for that. But I still do not see an advantage of SWAT2005 compared to more recent versions.*

Reply: Thanks for pointing out the incorrect descriptions. Here we want to explain the selection of SWAT 2005 used in this study more clearly.

Firstly, since the new versions of the SWAT model, i.e. SWAT 2009 and SWAT 2012, have removed the auto-calibration routing "automet.f" in the source codes, the modifications of these new version models then need to consider the coupling issues with the SWAT-CUP. However, the SWAT 2005 incorporates an internal auto-calibration routing, which makes it much easier to do model modifications.

Secondly, we have checked the revision history of the SWAT model and a previous study (Seo et al., 2014), model improvements were mainly focused on the water quality simulation. In this study, we think the

SWAT 2005 was generally adequate for flood simulation.

Finally, model modifications based on the SWAT 2005 could be also found in other studies (Dechmi et al., 2012;Jeong et al., 2010).

Overall, to justify the application of the SWAT2005, the original contents were replaced by following contents in P3, L31-P4, L6:

The source code of SWAT2005 has an internal auto-calibration module and such integrated design of model simulation and auto-calibration is easily manageable and modified since there is no need to couple external optimization algorithms. The accessible SWAT2009 (rev. 528) and SWAT2012 (rev. 664) have removed auto-calibration routines, however, an independent program SWAT-CUP (Abbaspour et al., 2007) is provided instead. Admittedly, many improvements have been made from the SWAT2005 to the latest SWAT2012. According to the SWAT model updates in (Seo et al., 2014), the major enhancements focused on the water quality modelling components, whereas the runoff modelling components in new SWAT versions were not so far different from those in the SWAT2005. This study was specific to the model modifications in runoff simulation, thus, the SWAT2005 was considered to be appropriate. Some other studies also modified the main hydrological processes based on the SWAT2005 version (Dechmi et al., 2012;Jeong et al., 2010).

*Comment: P.4., L.27: How can you consider 138 gauges for 21 subbasins using Thiessen average rainfall? It is unclear how spatial rainfall variability is included keeping in mind that the SWAT model is using one rainfall station per subbasin. And an accurate rainfall estimation might be crucial for flood forecasting. Please give more details on this method.*

Reply: Since we have delineated the catchment into 136 subbasins, the majority of the rainfall gauges were considered. The application of the rainfall data was added in P5, L9-16 as follow:

By default, SWAT structure allows only one rainfall input for each delineated sub-basin. Thus, sub-basins without available rainfall gauge would be automatically assigned the nearest one. For sub-basins with multiple rainfall gauges, Thiessen polygon method (Thiessen, 1911) was utilized to derive the rainfall input. Rainfall is the main driving force for hydrological models, and therefore accurate representation of spatially distributed rainfall is essential in hydrological modeling. (Cho et al., 2009) compared three different methods to incorporate spatially variable rainfall into the SWAT model and recommended the Thiessen polygon approach in catchments with high spatial variability of rainfall due to its robustness to catchment delineation.

*Comment: P.5, L.8: Again information is required how to consider „timely flood warning".*

Reply: This part has been deleted since this topic has been change to flood simulation.

*Comment: P.6, L. 21: A core term in calculating the time of concentration is the average slope length,*

*which is the average of the slope length within a subbasin. This parameter rather critical. Its spatial variability within a subbasin may increase with increasing catchment size. Thus, again this is a point which should be considered in the discussion of spatial heterogeneity. I assume that a larger number of subbasins and a better representation of the slope length might result in a more realistic modeling.*

Reply: Thanks for the comment. We have re-delineated the whole catchment into 136 sub-basins, the geographic features of sub-basins were presented in Table 2. We also want to state that the average slope length is a core term in calculating the overland flow according to Eq. (5). However, the final time of concentration in a sub-basin is dominated by the channel flow time, since the values of channel flow time are much greater than those of overland flow time

*Comment: P.7, L.16: Certainly, it is not required to calibrate 26 model parameters to model floods with the SWAT model. I recommend using less model parameters based on hydrological knowledge.*

Reply: Thanks for the good recommend. We agree that it is important to choose model parameters based on hydrological knowledge and watershed characteristics. For example, parameters relating to the snow melt were not included in the sensitivity analysis and model calibration due to the warm temperate in Huiahe River basin.

The original part was replaced by following contents in P8, L15-20:

It is highly recommended to identify the model parameters that can represent the hydrological characteristics of specific catchment before blindly applying sensitivity analysis. Based on the review of the SWAT model applications (Griensven et al., 2006;Cibin et al., 2010;Roth and Lemann, 2016) and the analysis of the SWAT model parameters, a total of 16 parameters related to the flow simulation in study area were involved in sensitivity analysis (see 错误!未找到引用源。) for daily simulation with the SWAT model. When it came to the event-based flood simulation with SWAT-EVENT model, additional distributed UH parameter $t_{\text{subadj}}$ (i.e., a total of 17 model parameters) were also considered.

*Comment: P.7, L.17: It is reasonable to use only the sensitive model parameters. However, it would be great to know how this decision was made. Thus, how were model parameters identified as sensitive?*

Reply: Thanks for the comment. To clearly explain how the sensitivity analysis works and how the sensitive parameters were identified, we added the following contents in P8, L6-25:

LH-OAT method firstly subdivides each parameter into $N$ stratums with a probability of $1/N$. Sampling points are randomly generated so that one parameter is sampled only once at each strata. Then, the local sensitivity of a parameter at one sampling point is calculated as:

$$S_{ij} = 200g \left| \frac{\left[ y(\theta_1,...,\theta_i + \Delta_i,...,\theta_P) - y(\theta_1,...,\theta_P) \right] / \left[ y(\theta_1,...,\theta_i + \Delta_i,...,\theta_P) + y(\theta_1,...,\theta_P) \right]}{\Delta_i} \right|$$

(7)

where $S_{ij}$ is the partial effect of parameter $\theta_i$ at the LH sampling point $j$; $y$ is the model output (or objective function); $\Delta_i$ is the perturbation of parameter $\theta_i$ and $P$ is the number of parameters. The final sensitivity index $S_i$ for the parameter $\theta_i$ is derived by averaging these partial effects of each loop for all LH points (i.e., $N$ loops). The greater the $S_i$, the more sensitive the model response is to that particular parameter.

It is highly recommended to identify the model parameters that can represent the hydrological characteristics of specific catchment before blindly applying sensitivity analysis. Based on the review of the SWAT model applications (Griensven et al., 2006;Cibin et al., 2010;Roth and Lemann, 2016) and the analysis of the SWAT model parameters, a total of 16 parameters related to the flow simulation in study area were involved in sensitivity analysis (see 错误!未找到引用源。) for daily simulation with the SWAT model. When it came to the event-based flood simulation with SWAT-EVENT model, additional distributed UH parameter $t_{subadj}$ (i.e., a total of 17 model parameters) were also considered. For both models, the objective function $y$ in Eq. (7) represented the residual sum of squares of stream flow between the simulated set and the measured set. Specifically, sensitivity analysis of the SWAT model was conducted not only for long-term period, but also for the same flood periods as the SWAT-EVENT simulation. According to the sensitivity ranks of $S_i$ results, the upper-middle ranking parameters would be used for the calibration procedure, while the values of the other parameters were set to their default values.

*Comment: P.7, L.21: I cannot follow the justification for an automatic calibration. Certainly it is not possible to calibrate 26 parameters manually in a realistic time period. However, it is certainly possible to reduce the number of parameters to an acceptable number.*

Reply: Thanks for the comment. The original part was deleted, instead, the following content was added in P8, L27-28:

Before effectively applying a hydrological model, a calibration process aims to estimate the model parameters that minimize the errors between the observed and simulated results is usually necessary.

*Comment: P.8, L.21: It would be interesting to see a sensitivity analysis specific for SWAT-EVENT.*

Reply: Sensitivity analysis result was added in Table 3, and the corresponding descriptions were added in P10, L14-P5, L3.

*Comment: P.9, L.15: It is rather strange to highlight that the performance is very good according to ENS, but to ignore that the performance is only satisfactory according to PBIAS refered to the same*

*reference. Please consider all thresholds or none of them.*

Reply: Since the catchment was further divided into 136 sub-basins, model calibration and validation were re-worked. The optimal model parameters were listed in Table 4 and the model performance was presented in Table 5. All the three statistical indicators were evaluated according to performance ratings for evaluation statistics recommended by (Moriasi et al., 2007). The original part was replaced by the following content in P11, L6-13:

The $E_{NS}$ value is 0.76 for the calibration period and 0.80 for the validation period. These two values of the daily $E_{NS}$ both exceed 0.75, which is considered to be "very good" according to performance ratings for evaluation statistics recommended by (Moriasi et al., 2007). The daily $R_{SR}$ values are 0.49 and 0.44 for the calibration and validation, respectively, indicating that the root mean square error values are less than half the standard deviation of measured data, i.e. the "very good" model performances suggested by (Moriasi et al., 2007). The SWAT model overestimates the streamflow by 5.72 % for calibration while underestimating the streamflow by 8.38 % for validation. The calculated results of $P_{BIAS}$ in 错误!未找到引用源。 also attain the "very good" rating.

*Comment: P.12, L.18-21: The selection of objective functions is certainly a trade-off in optimizing different parts of the hydrograph. A focus on high flows might lead to lower performance for normal or low flows. It should be made clear that the selection of the objective function was made to fulfill the goals of the study and that it was accepted to have a lower performance in other phases of the hydrograph.*

Reply: Thanks for the comment. We added the following illustration in P14, L24-30:

As Coustau et al. (2012) declared, event-based models were very convenient for operational purposes, if the initial wetness state of the catchment would be known with good accuracy. Although the continuous modeling approach used in this study was not the perfect solution for the determination of the catchment antecedent conditions, it was still an effective method as the preparatory preparation for the simulation of the SWAT-EVENT model due to the good goodness-of-fit in 错误!未找到引用源。 and 错误!未找到引用源。. Since the goal of this research was to ascertain the applicability of the newly developed SWAT-EVENT model on event-based flood simulation, it was accepted to have a lower performance in calculating the antecedent conditions.

*Comment: Table A1: Several parameters are certainly not relevant in modelling flood peaks such as BIOMIX, SOL_ALB, TLAPS, etc. Please reduce the number of parameters based on hydrological knowledge in advance.*

Reply: Table A1 was changed to Table 3 and the number of model parameters was reduced based on

hydrological knowledge and watershed characteristics.

*Comment: Table A1: Please explain the meaning of -25 and 25 for snow melt temperature (SMTMP).*

Reply: Table A1 was changed to Table 3 and explanatory notes were added below the Table 3.

*Comment: Table A1: The slope length is used first to describe the geographical features of the subbasins and then it is calibrated. Is this approach valid?*

Reply: It is not valid to calibrate the slope length since we think this parameter was obtained from the DEM. Thus, slope length parameter was deleted from Table 3.

*Comment: Table A2: The differences in model parameters between the SWAT model and the SWAT-EVENT version is very high. From hydrological point of view, it does not make sense to use very different parameter sets of the same model in the same catchment. This means that the hydrological processes are not well represented/understood.*

Reply: Table A2 was change to Table 4. Here we respectfully argue that the SWAT model and the SWAT-EVNET model were not exactly the same. The SWAT-EVENT was simulating at a sub-daily time scale while the SWAT model was simulating at daily time scale. This did not mean that the only difference between the two models was the model input. As stated by (Jeong et al., 2010), modifications from daily time scale to sub-daily time scale have changed many hydrological processes (i.e. calculation of infiltration, and surface runoff lag and channel routing) in the original SWAT model. What's more, the UH routing method was only used for sub-daily simulation. The daily simulation with the SWAT model presented in this study did not take into account the UH routing method. Therefore, we think it was reasonable to have differences in model parameters between the SWAT model and the SWAT-EVENT version.

*Comment: Table 4: All flood events are summer events. Do you really need to consider snow model parameters?*

Reply: Model parameters used in sensitivity analysis and calibration were refreshed in Table3 and Table 4.

*Comment: Figure 2a: can be removed since land use and soil types are not discussed in detail.*

Reply: As reviewer suggested, Figure 2 was removed.

**3. Responses to other comments**

*Comment: P.2., L. 16: Small letters for Beven and Kirkby.*

Reply: This part has been removed due to the comments above.

*Comment: P.2., L.22: HRU.*

Reply: We have made correction in P2, L30.

*Comment: P.6, Eq.2-6: Not all terms of these equations are clearly visible.*

Reply: We are very sorry for our incorrect writing. We have fixed all the unclearly equations.

*Comment: P.11, L.5: concluded*

Reply: We have made correction.

*Comment: Figure 8: basin instead of bsin (x-axis)*

Reply: We have made correction.

[revised manuscript text omitted]

5   **Table 3 Parameters and parameter ranges used in sensitivity analysis and the final ranks of sensitivity analysis results.**

| Parameter | Definition | Lower bound | Upper bound | Daily simulation with SWAT model | | Event-based sub-daily simulation with SWAT-EVENT model |
|---|---|---|---|---|---|---|
| | | | | Long-term period | Flood period | |
| ALPHA_BF | Baseflow alpha factor (days). | 0 | 1 | 4 | 3 | 4 |
| BLAI | Maximum potential leaf area index. | 0 | 1 | 10 | 8 | 15 |
| CANMX | Maximum canopy storage (mm). | 0 | 10 | 11 | 11 | 12 |
| CH_K2 | Effective hydraulic conductivity in main channel alluvium (mm/hr). | 0 | 150 | 5 | 5 | 11 |
| CH_N2 | Manning's "n" value for the main channel. | 0.01 | 0.3 | 1 | 1 | 1 |
| CN2 [a] | Initial SCS runoff curve number for moisture condition II. | -25 | 25 | 3 | 4 | 2 |
| EPCO | Plant uptake compensation factor. | 0 | 1 | 12 | 12 | 16 |
| ESCO | Soil evaporation compensation factor | 0 | 1 | 6 | 6 | 17 |

| Parameter | Description | | | | | |
|---|---|---|---|---|---|---|
| GW_DELAY | Groundwater delay time (days). | 0 | 20 | 15 | 13 | 10 |
| GW_REVAP [b] | Groundwater "revap" coefficient. | -0.036 | 0.036 | 14 | 14 | 14 |
| GWQMN | Threshold depth of water inthe shallow aquifer required for return flow to occur (mm). | 0.01 | 100 | 8 | 9 | 7 |
| REVAPMN [b] | Threshold depth of water in the shallow aquifer for "revap" or percolation to the deep aquifer to occur (mm). | -100 | 100 | 16 | 16 | 13 |
| SOL_AWC [a] | Available water capacity of the soil layer (mm /mm ). | -30 | 30 | 7 | 7 | 5 |
| SOL_K [a] | Saturated hydraulic conductivity (mm/hr). | -50 | 50 | 13 | 15 | 8 |
| SOL_Z [a] | Depth from soil surface to bottom of layer (mm). | -30 | 30 | 9 | 10 | 6 |
| SURLAG | Surface runoff lag coefficient. | 0 | 20 | 2 | 2 | 9 |
| $t_{subadj}$ [a] | Sub-basin level UH parameter (h) | -50 | 50 | | | 3 |

[a] These parameters are varied by multiplying a ratio (%) within the range.

[b] These parameters are varied by adding or subtracting a value within the range.

批注 [L34]: Table 3 was added to show the model parameter used in this study.

**Table 4 Calibrated parameter values for the SWAT model and the SWAT-EVENT model.**

批注 [L35]: Table 4 was added to show the optimal parameter values.

| | Daily simulation with the SWAT model | | Event-based simulation with the SWAT-EVENT model | |
|---|---|---|---|---|
| Parameter | Value for long-term period calibrating | Value for flood period calibrating | Parameter | Value |
| CH_N2 | 0.10 | 0.19 | CH_N2 | 0.03 |
| SURLAG | 1.84 | 2.40 | CN2 | 24.60 |
| CN2 | 15.98 | 20.68 | $t_{subadj}$ | -10.40 |
| ALPHA_BF | 0.84 | 0.75 | SOL_Z | -7.91 |
| CH_K2 | 109.90 | 54.00 | GWQMN | 0.28 |
| ESCO | 0.94 | 1.00 | SOL_AWC | -29.71 |
| SOL_AWC | -18.01 | -9.26 | ALPHA_BF | 0.88 |
| | | | SOL_K | -48.84 |

**Table 5 SWAT model performance statistics for long-term period calibrating and flood period calibrating..**

| Statistical indicator | Long-term period calibrating | | Flood period calibrating | |
|---|---|---|---|---|
| | Calibration | Validation | Calibration | Validation |
| $E_{NS}$ | 0.76 | 0.80 | 0.78 | 0.81 |
| $R_{SR}$ | 0.49 | 0.44 | 0.48 | 0.44 |
| $P_{BIAS}$ (%) | 5.72 | -8.38 | 5.27 | -6.10 |

批注 [L36]: Table 5 was added to show the daily simulation results with the SWAT model.

批注 [L37]: Table 6 was added to show the sub-daily flood simulation with the SWAT-EVENT model.

[revised manuscript text omitted]

批注 [L43]: Figure 9 was added to illustrate the SWAT-EVENT performance for the flood peak simulation.

[Figure]

**Figure 10 Box plots of ENS values for the SWAT-EVENT model results for sub-basin level UH parameters and basin level UH parameters.**

批注 [L44]: Figure 10 was modified due to the re-calibration of the SWAT-EVENT model.

---

## Author Response (AR3)

**Response to interactive comment**

We heartily appreciate the reviewer's assessments on this study and the valuable suggestions provided to improve this manuscript. We also would like to thank the reviewer for the recognition of previous revises of this manuscript. We have studied comments carefully and have made corrections which we hope meet with the final approval. We hereby provide our point by point responses how the comments by editor and referee will be addressed in the revised manuscript.

**1. Responses to editor**

*Comment: I also had a look at the revised paper and I would like to ask you, to revise some of the figures to ensure better readability. The figure captions are very small in Fig 5,6,7,8. You should also remove the border in Figure 9 - this in untypical for the HESS style.*

Reply: Thanks for the comments. We have enlarged the figure captions in Fig 5, 6, 7 and 8. And the border in Figure 9 has been removed.

**2. Responses to referee**

*Comment: Tab. 3: The baseflow alpha factor is ranked on the third position in the flood period (compared to position 4 in the long-term period). This is really a surprising result which needs to be discussed. From hydrological point, it seems to that the process representation is weak in the case that baseflow controls flood. This is moreover critical since the curve number CN2 as the most relevant model parameter for high flows in many SWAT studies is only ranked on position 4. Thus, to summarize it: Based on the results of the sensitivity analysis, a baseflow parameter is more relevant for floods than a parameter regulating surface runoff and infiltration. This unexpected result needs to be explained.*

Reply: Thanks for the comments. It is really true as reviewer suggested that the unexpected sensitivity results of the CN2 and ALPHA_BF need to be explained.

In a previous study, Bondelid et al. (2010) analyzed the sensitivity of SCS models to the variation of CN2 parameter, and found that sensitivity results of CN2 were influenced by rainfall. More specifically, the proportional change in SCS model output resulting from a unit change in CN2 decreased as the rainfall and increased. As confirmed by Cibin et al. (2010), the parameter CN2 would become less sensitive to wet year simulation and high flow simulation for the SWAT model. The following explanation was added in P10, L19 - 28 to explain the reduced sensitivity rank of parameter CN2:

Unexpectedly, compared to the long-term analysis, the initial SCS runoff curve number (CN2) shows less effect on streamflow output during flood period, whereas the groundwater parameter ALPHA_BF becomes more sensitive to high streamflow regime. As declared by Bondelid et al. (2010), the effects of CN2 variation on surface runoff yield decreased as the rainfall increased, especially for the larger storm events. Bondelid et al. (2010) further explained that the proportion of the rainfall that went into initial abstraction and infiltration decreased along with the increasing of rainfall, so the proportional changes in surface runoff associated with a unit change in CN2 would decrease. Furthermore, from a previous sensitivity study with the SWAT model (Cibin et al., 2010), the parameter CN2 in wet year simulation was found to be less important than that in entire simulation, and the greatest sensitivity index of CN2 was found in low flow. Thus, there is reason to believe, the sensitivity ranking of CN2 would be reduced when it comes to flood period analysis in Table 3.

In addition, we added the following explanation in P10, L28 - L30 to illustrate the performance of parameter ALPHA_BF:

Instead, in this process, the model output changes resulting from the perturbation of parameter ALPHA_BF would be more prominent, as there is more water recharging the shallow aquifer, and meanwhile the parameter ALPHA_BF strongly influences groundwater response to changes in recharge (Sangrey, 1984).

**Comment: Tab. 4: Please explain the final values for CN2. Have you added these values to the initial CN2?**

Reply: Thanks for the comment. For parameter like CN2, the distributed values are varied according to a relative change that maintains their spatial relationship. To clearly explain the final optimal results, we have added explanatory notes below the Table 4 as follows:

The final values of these parameters are derived by multiplying the percentage change (%) based on their default values. Parameter CN2 with the calibrated value of 15.98, for example, means that the default values are multiplied by (1 + 15.98 %) to obtain the optimal results.

**Comment: Tab. 4: The values for ALPHA_BF are rather high which means that the baseflow reacts fast. Could you refer this to the catchment characteristics (by also considering my first comment).**

Reply: Thanks for the comment. Parameter ALPHA_BF is a direct index of ground water flow response to changes in recharge (Kiniry et al., 2005). Theoretically this parameter should be related to the characteristics of shallow aquifer. According to the geological survey of the Huaihe River basin, we found that the upper Huaihe River basin had good drainage condition. Thus, the content was added in P11, L1 - L2:

Considering that the shallow aquifer in the Huaihe River basin has good drainage condition (Zuo et al., 2006), a relatively

high value of ALPHA_BF would be expected in this study.

*Comment: Tab. 6: Please check the layout in the table header.*

Reply: We have made the table header well-formed.

[revised manuscript text omitted]
} (Q_{obs}(i) - Q_{sim}(i))^2}}{\sqrt{\sum_{i=1}^{n} (Q_{obs}(i) - \overline{Q_{obs}})^2}} \tag{9}$$

$$P_{BIAS} = \left[ \frac{\sum_{i=1}^{n} (Q_{obs}(i) - Q_{sim}(i)) \cdot 100}{\sum_{i=1}^{n} Q_{obs}(i)} \right] \tag{10}$$

where $Q_{obs}(i)$ is the $i$ th observed streamflow (m$^3$ s$^{-1}$); $Q_{sim}(i)$ is the $i$ th simulated streamflow (m$^3$ s$^{-1}$); $n$ is the length of the time series.

**3.3.3 Event-based sub-daily calibration and validation with the SWAT-EVENT model**

In this study, the SWAT-EVENT model employed the same built-in automatic calibration subroutine as the SWAT model did. Sub-daily simulations with the SWAT-EVENT model were conducted within the same time span as the daily simulation, with a primary focus on the flood season with a series consisting of 24 flood events, two-thirds of which were utilized for the calibration while the rest were used for validation. Preferential implementation was applied to daily calibration from which the antecedent conditions were extracted.

$E_{NS}$, relative peak discharge error ($E_{RP}$), relative peak time error ($E_{RPT}$) and relative runoff volume error ($E_{RR}$) were selected as the performance evaluation statistics for the flood event simulations to comply with the Accuracy Standard for Hydrological

Forecasting in China (MWR, 2008). $E_{RP}$, $E_{RPT}$, and $E_{RR}$ are specific indicators used to indicate whether the accuracies of the simulations reach the national standard (MWR, 2008). They are considered to be sufficiently qualified when the absolute values are less than 20 %, 20 % and 30 %, respectively.

**4 Results**

**4.1 Sensitivity analysis results**

Sensitivity results for daily simulation with the SWAT model are listed in Table 3. The sensitivity rank for a single parameter shows tiny differences between the two types of analysis period for SWAT simulation, with the changes in all parameter ranks less than 3. According to a previous study (Cibin et al., 2010), the sensitivity of SWAT parameters was proved to vary in low, medium and high streamflow regimes. The long-term period analysis in Table 3 consists of different flow regimes, but presents almost the same sensitivity ranks as the flood period case, indicating that the high streamflow would dominate the sensitivity results in the long-term period analysis. Unexpectedly, compared to the long-term analysis, the initial SCS runoff curve number (CN2) shows less effect on streamflow output during flood period, whereas the groundwater parameter ALPHA_BF becomes more sensitive to high streamflow regime. As declared by Bondelid et al. (2010), the effects of CN2 variation on surface runoff yield decreased as the rainfall increased, especially for the larger storm events. Bondelid et al. (2010) further explained that the proportion of the rainfall that went into initial abstraction and infiltration decreased along with the increasing of rainfall, so the proportional change in surface runoff associated with a unit change in CN2 would decrease. Furthermore, from a previous sensitivity study with the SWAT model (Cibin et al., 2010), the parameter CN2 in wet year simulation was found to be less important than that in entire simulation, and the greatest sensitivity index of CN2 was found in low flow. Thus, there is reason to believe, the sensitivity ranking of CN2 would be reduced when it comes to flood period analysis in Table 3. Instead, in this process, the model output changes resulting from the perturbation of parameter ALPHA_BF would be more prominent, as there is more water recharging the shallow aquifer, and meanwhile the parameter ALPHA_BF strongly influences groundwater response to changes in recharge (Sangrey, 1984). Considering that the shallow aquifer in the Huaihe River basin has good drainage condition (Zuo et al., 2006), a relatively high value of ALPHA_BF would be expected in this study. Generally, the identified 7 sensitive parameters of the daily SWAT model cover multiple main hydrological processes, i.e. channel routing (CH_N2 and CH_K2), runoff (SURLAG and CN2), groundwater (ALPHA_BF), evaporation (ESCO) and soil water (SOL_AWC), not only for long-term period but also for flood period. According to Table 3, it is clear that both the year-round streamflow and the high streamflow are most sensitive to CH_N2 due to its top sensitivity rank.

Table 3 also presents the sensitivity results for event-based flood simulation with SWAT-EVENT model at sub-daily time scale. Sensitivity of some parameters differs widely from its performance in flood period analysis with SWAT model at daily time scale. The sensitivity ranks of BLAI, CH_K2, ESCO, SOL_K, and SURLAG have changed more than 5, which could be caused by the differences in hydrological simulation between the SWAT model and the SWAT-EVENT model. It is

批注 [u1]: The explanation for the unexpected sensitivity results of the CN2 and ALPHA_BF was added.

批注 [u2]: The justification of the parameter ALPHA_BF was added.

[revised manuscript text omitted]

Zuo, Z., Wang, X., Luo, W., Wang, F., and Guo, S.: Characteristics on Aquifer of the Quaternary system in Huai River Basin(Henan Section), Ground Water, 2006.

**Table 1 SWAT model input data and sources for the Wangjiaba (WJB) catchment.**

| Data type | Resolution | Source | Description |
|---|---|---|---|
| DEM | 90m×90m | http://srtm.csi.cgiar.org/ | Digital Elevation Model |
| Land use | 1km×1km | http://www.landcover.org/ | Land use classification |
| Soil | 30 arc-second | http://www.fao.org/soils-portal/soil-survey/soil-maps-and-databases/harmonized-world-soil-database-v12/en/ | Soil type classification and characterization of soil parameters |
| Global weather data | 30 stations | https://globalweather.tamu.edu/ | Relative humidity, wind speed, solar radiation and the minimum and maximum air temperatures |
| Observed rainfall | 138 gauges | Hydrologic Bureau of Huaihe River Commission | Daily data: 1991-2010; sub-daily data: flood period during 1991-2010 |
| Observed streamflow | 1 gauges | Hydrologic Bureau of Huaihe River Commission | Wangjiaba station, daily data for 1991-2010, sub-daily data for flood period during 1991-2010 |

**Table 2 Geographic features of sub-basins for the Wangjiaba (WJB) catchment.**

| | $L_{slp}$ | $S_{sub}$ | $L$ | $A$ | $S_{ch}$ | $t_{ov}$ | $t_{ch}$ | $t_c$ |
|---|---|---|---|---|---|---|---|---|
| | (m) | (m/m) | (km) | (km$^2$) | (m/m) | (h) | (h) | (h) |
| Minimum | 28.46 | 0.01 | 0.71 | 0.09 | 0.000 | 0.14 | 0.13 | 1.37 |
| Maximum | 121.95 | 0.22 | 96.83 | 879.16 | 0.024 | 2.42 | 33.06 | 34.18 |
| Average | 100.42 | 0.04 | 37.44 | 221.88 | 0.005 | 0.91 | 6.03 | 6.94 |
| CV | 0.29 | 1.28 | 0.52 | 0.74 | 1.18 | 0.37 | 0.91 | 0.81 |

5  **Table 3 Parameters and parameter ranges used in sensitivity analysis and the final ranks of sensitivity analysis results.**

| Parameter | Definition | Lower bound | Upper bound | Daily simulation with SWAT model | | Event-based sub-daily simulation with SWAT-EVENT model |
|---|---|---|---|---|---|---|
| | | | | Long-term period | Flood period | |
| ALPHA_BF | Baseflow alpha factor (days). | 0 | 1 | 4 | 3 | 4 |
| BLAI | Maximum potential leaf area index. | 0 | 1 | 10 | 8 | 15 |
| CANMX | Maximum canopy storage (mm). | 0 | 10 | 11 | 11 | 12 |
| CH_K2 | Effective hydraulic conductivity in main channel alluvium (mm/hr). | 0 | 150 | 5 | 5 | 11 |
| CH_N2 | Manning's "n" value for the main channel. | 0.01 | 0.3 | 1 | 1 | 1 |
| CN2 [a] | Initial SCS runoff curve number for moisture condition II. | -25 | 25 | 3 | 4 | 2 |
| EPCO | Plant uptake compensation factor. | 0 | 1 | 12 | 12 | 16 |
| ESCO | Soil evaporation compensation factor | 0 | 1 | 6 | 6 | 17 |

| | | | | | | |
|---|---|---|---|---|---|---|
| GW_DELAY | Groundwater delay time (days). | 0 | 20 | 15 | 13 | 10 |
| GW_REVAP [b] | Groundwater "revap" coefficient. | -0.036 | 0.036 | 14 | 14 | 14 |
| GWQMN | Threshold depth of water inthe shallow aquifer required for return flow to occur (mm). | 0.01 | 100 | 8 | 9 | 7 |
| REVAPMN [b] | Threshold depth of water in the shallow aquifer for "revap" or percolation to the deep aquifer to occur (mm). | -100 | 100 | 16 | 16 | 13 |
| SOL_AWC [a] | Available water capacity of the soil layer (mm /mm ). | -30 | 30 | 7 | 7 | 5 |
| SOL_K [a] | Saturated hydraulic conductivity (mm/hr). | -50 | 50 | 13 | 15 | 8 |
| SOL_Z [a] | Depth from soil surface to bottom of layer (mm). | -30 | 30 | 9 | 10 | 6 |
| SURLAG | Surface runoff lag coefficient. | 0 | 20 | 2 | 2 | 9 |
| $t_{subadj}$ [a] | Sub-basin level UH parameter (h) | -50 | 50 | | | 3 |

[a] These parameters are varied by multiplying a ratio (%) within the range.

[b] These parameters are varied by adding or subtracting a value within the range.

**Table 4 Calibrated parameter values for the SWAT model and the SWAT-EVENT model.**

| | Daily simulation with the SWAT model | | Event-based simulation with the SWAT-EVENT model | |
|---|---|---|---|---|
| Parameter | Value for long-term period calibrating | Value for flood period calibrating | Parameter | Value |
| CH_N2 | 0.10 | 0.19 | CH_N2 | 0.03 |
| SURLAG | 1.84 | 2.40 | CN2 [a] | 24.60 |
| CN2 [a] | 15.98 | 20.68 | $t_{subadj}$ [a] | -10.40 |
| ALPHA_BF | 0.84 | 0.75 | SOL_Z [a] | -7.91 |
| CH_K2 | 109.90 | 54.00 | GWQMN | 0.28 |
| ESCO | 0.94 | 1.00 | SOL_AWC [a] | -29.71 |
| SOL_AWC [a] | -18.01 | -9.26 | ALPHA_BF | 0.88 |
| | | | SOL_K [a] | -48.84 |

[a] The final values of these parameters are derived by multiplying the percentage change (%) based on their default values. Parameter CN2 with the calibrated value of 15.98, for example, means that the default values are multiplied by (1 + 15.98 %) to obtain the optimal results.

批注 [u3]: The explanation for the final optimal results of the distributed parameters was added.

[revised manuscript text omitted]